# Cargo surface fluidity can reduce inter-motor mechanical interference, promote load-sharing and enhance processivity in teams of molecular motors

**Niranjan Sarpangala**, **Ajay Gopinathan** *

Department of Physics, and Center for Cellular and Biomolecular Machines, University of California, Merced, California, United States of America

* agopinathan@ucmerced.edu

**Data Availability Statement:** All relevant data are within the manuscript and its Supporting

## Abstract

In cells, multiple molecular motors work together as teams to carry cargoes such as vesicles and organelles over long distances to their destinations by stepping along a network of cytoskeletal filaments. How motors that typically mechanically interfere with each other, work together as teams is unclear. Here we explored the possibility that purely physical mechanisms, such as cargo surface fluidity, may potentially enhance teamwork, both at the single motor and cargo level. To explore these mechanisms, we developed a three dimensional simulation of cargo transport along microtubules by teams of kinesin-1 motors. We accounted for cargo membrane fluidity by explicitly simulating the Brownian dynamics of motors on the cargo surface and considered both the load and ATP dependence of single motor functioning. Our simulations show that surface fluidity could lead to the reduction of negative mechanical interference between kinesins and enhanced load sharing thereby increasing the average duration of single motors on the filament. This, along with a cooperative increase in on-rates as more motors bind leads to enhanced collective processivity. At the cargo level, surface fluidity makes more motors available for binding, which can act synergistically with the above effects to further increase transport distances though this effect is significant only at low ATP or high motor density. Additionally, the fluid surface allows for the clustering of motors at a well defined location on the surface relative to the microtubule and the fluid-coupled motors can exert more collective force per motor against loads. Our work on understanding how teamwork arises in cargo-coupled motors allows us to connect single motor properties to overall transport, sheds new light on cellular processes, reconciles existing observations, encourages new experimental validation efforts and can also suggest new ways of improving the transport of artificial cargo powered by motor teams.

## Author summary

In cells, multiple molecular motors work together as teams to carry cargoes such as vesicles and organelles over long distances to their destinations by stepping along a network

information files and github repository (https://github.com/nsarpangala/lipid-cargo-transport).

**Funding:** This work was supported by the National Science Foundation (NSF-DMS-1616926 to AG) and NSF-CREST: Center for Cellular and Biomolecular Machines at UC Merced (NSF-HRD-1547848 to AG). AG and NS also acknowledge partial support from the NSF Center for Engineering Mechanobiology grant CMMI-154857 and computing time on the Multi-Environment Computer for Exploration and Discovery (MERCED) cluster at UC Merced (NSF-ACI-1429783). NS acknowledges Graduate Student Opportunity Program Fellowship from the University of California, Merced. The funders played no role in the study design, data collection and analysis, decision to publish, or preparation of the manuscript.

**Competing interests:** The authors have declared that no competing interests exist.

of protein filaments. How do motors that interfere with each other when coupled rigidly in laboratory experiments, function well as teams within cells? In this paper, we show, using computer simulations, that the fluid surfaces of cellular cargo reduce the mechanical interference between motors allowing better load sharing and increased duration of motors on the filament, thereby increasing the distance over which they can carry cargo. We additionally found that motors pull cargo closer to the filament increasing the attachment rates of other unbound motors. These effects act synergistically with an increased availability of motors due to fluidity to further increase travel distances. Fluid surfaces also allow clustering of the motors and increased collective force against load. Our work on understanding how teamwork arises in mechanically coupled motors sheds new light on cellular processes, reconciles existing observations, encourages new experimental validation efforts and can also suggest new ways of improving transport of artificial cargo powered by motor teams.

## Introduction

In eukaryotic cells, the intracellular transport of material between various organelles and the cellular and nuclear membranes is critical for cellular function and is an active process facilitated by the consumption of energy [1, 2]. A variety of motor proteins including myosins, kinesins and dyneins convert chemical energy from ATP hydrolysis into directed stepping along cytoskeletal protein filaments such as actin filaments and microtubules [1, 3]. These motors move in different directions along the filaments typically carrying lipid bilayer vesicles packed with proteins and signaling molecules or even membrane bound organelles such as Golgi and mitochondria [1, 2]. Defects in motor function can impair normal cell functioning and lead to a variety of pathologies including neurodegenerative diseases like Alzheimer's disease and ALS [4–9]. Given its importance *in vivo*, there has been a lot of work done over the years in understanding motor function in the transport context ranging from single molecule studies that elucidate the detailed mechanistic workings of motors [10–17] to the functioning of multiple motors and their co-ordination [18–27]. Several theoretical and *in vitro* experimental studies [28–32] indicate that the collective behavior of motors is critically influenced by their coupling to each other resulting in observable effects in transport speeds and run lengths. For example, it is known that non-cooperative kinesin motors coupled together by a rigid cargo interfere with each other's functioning leading to enhanced detachments and lowered run lengths [28–30]. How then are *in vivo* cargoes such as membrane-bound vesicles and organelles typically carried over long distances by teams of kinesin motors [33, 34]? It is possible that a fluid membrane simply makes more motors available for binding at the filament either by dynamic clustering of motors [35, 36] or membrane induced clustering of motors into micro-domains [37, 38] thereby enhancing processivity. Alternatively or in addition, there could be other coupling-dependent physical mechanisms that directly affect single motor functioning and thereby promote teamwork. Studies on transport *in vivo* could address these questions directly, but the environmental complexity makes it difficult to disentangle fundamental phenomena of physical origin from the effects of other regulatory mechanisms [39]. *In vitro* systems, which offer cleaner insights, include microtubule gliding assays on flat bilayers with embedded motors [35, 40–42], motor driven nanotube extraction from vesicles [43] and more recently even membrane covered beads carried by kinesin motors [44]. Studies of these systems and related models have generated a variety of interesting and sometimes conflicting results on cargo run lengths and velocities in the presence of membrane fluidity

[36, 40, 44, 45]. These results are also obtained under different conditions of cargo geometry, motor density, cargo surface fluidity and environmental factors like ATP concentration. In order to reconcile them, we need an understanding of the relative importance and interplay of these factors and their effects on the number of engaged motors and the loads they experience, which in turn influences their collective speed and processivity. More importantly, to uncover any physical mechanisms present in teams of motors that directly affect single motor functioning and thereby enhance teamwork, we need to be able to monitor both single motor dynamics and collective cargo transport simultaneously.

Here, we used Brownian dynamics simulations to accomplish this goal. Briefly, we model the transport of a spherical cargo with a fluid surface carried by multiple kinesin-1 motors along a microtubule (MT) (see Fig 1 and details in Materials and methods). We explicitly accounted for the movement of the attachment points of the motors on the cargo surface ($A_i$ in Fig 1c) by allowing free diffusion for unbound motors and diffusion biased by exerted

### (a) Rigid cargo          (b) Lipid cargo

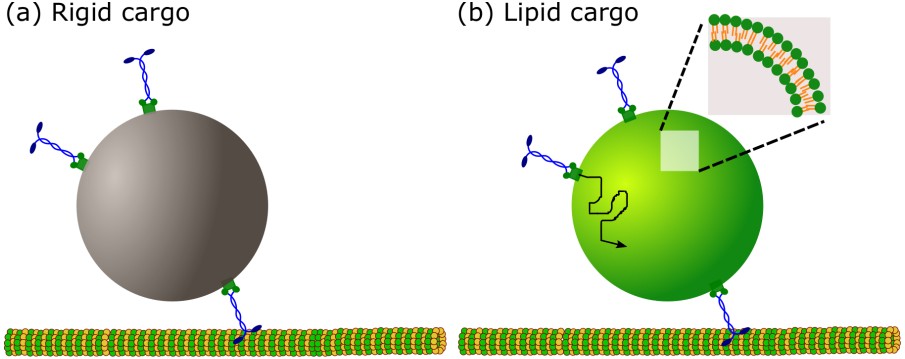

### (c) Illustration of the computation of motor forces

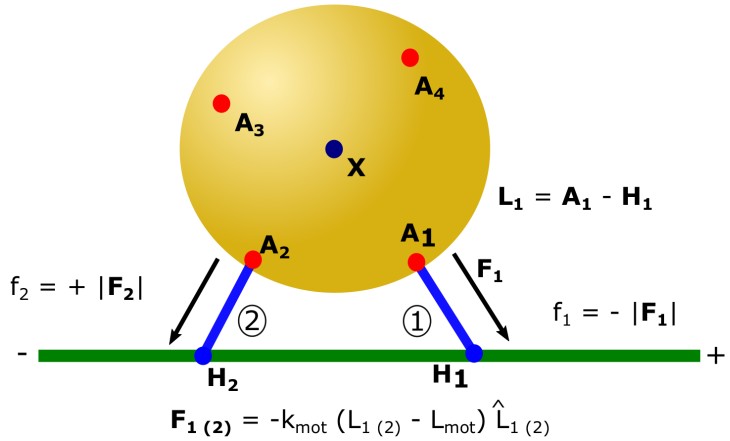

$$F_{1(2)} = -k_{mot}(L_{1(2)} - L_{mot})\hat{L}_{1(2)}$$

**Fig 1. Schematics of cargoes considered in the study.** **(a)** Rigid cargo (membrane-free cargo). Molecular motors are permanently attached to random locations on the cargo surface. **(b)** Fluid/Lipid cargo (membrane-enclosed cargo). Molecular motors diffuse on the cargo surface. Inset of Fig 1b The lipid bilayer that forms the fluid cargo surface. **(c)** Schematic that shows how we compute the forces generated by motors. $X$ is the center of mass of the cargo, $A_i$ and $H_i$ represent motor anchor and motor head positions. Motors exert spring like forces only when their lengths $L_i$ exceed their rest length $L_{mot}$. Here, motor 1 experiences a hindering load ($f_1$ is negative), while motor 2 experiences an assistive force ($f_2$ is positive).

forces (spring forces $F_i$ in Fig 1c) for bound motors, with cargo surface fluidity determining the diffusion constant. We also accounted for the force and ATP dependence of the bound motor's off-rate and stepping rate. The spherical cargo's position was then updated by Brownian dynamics depending on the net force exerted by all bound motors. We also incorporated the rotational diffusion of the cargo due to the torque from motor forces and thermal fluctuations. Our model is overall quite similar to the Brownian dynamics model of Bovyn *et. al.* [36] which also implemented overdamped Langevin dynamics of motors on an undeformable, spherical lipid cargo surface. While there are minor differences between our models such as in the details of the load dependence of the motor unbinding rates and the incorporation of ATP, we do not expect significant quantitative differences. Indeed, measurements of the same quantities such as run lengths and single motor binding times are consistent between our studies. The major difference with our work lies in the focus of their study which was the effect of surface fluidity on the availability of motors for both clustered and dispersed motors and how this influenced the trade-off between binding rates from solution and run length. While we also consider the availability of motors, a major focus in this paper is on the effect of fluidity at the single motor level. This includes effects on the mechanical interference between motors which affects their load-sharing and thereby single motor processivity as well as collective force generation. Another difference is that we specifically consider the effects of varying ATP concentration. Finally, we study the relative importance of these effects and how they act synergistically at the level of the collective to influence cargo transport.

Specifically, our model enabled us to focus on the dynamics of the motors as they diffused on the surface, their binding and unbinding from the filament and the forces experienced by individual motors, as a function of motor diffusion constant, motor density and ATP concentration. We connected these characteristics at the individual motor level to the effect of surface fluidity (inverse viscosity proportional to the diffusion constant) on the collective behavior of motors at the cargo level by analyzing transport properties such as the average number of bound motors, collective force generated against load, speed and distance traveled (run length). We uncovered two salient features, the reduction of interference and the cooperative increase of on-rates. Our simulations showed explicitly that surface fluidity leads to the reduction of negative mechanical interference between kinesins, characterized by lower forces on individual motors and a significant drop in antagonistic forces between motors. This allows teams of fluid-coupled motors to more fully exploit load sharing without inter-motor interference. This decreases single motor off-rates and increases processivity. Our simulations also showed that the fluid surface allows for the clustering of motors at a well defined location on the surface relative to the microtubule and that the fluid-coupled motors can exert more collective force per motor against loads. Interestingly, increasing numbers of bound motors pull the cargo closer to the microtubule, increasing the on-rates of unbound motors, resulting in a cooperative increase in bound motor numbers that depends on 3D cargo geometry. Surface fluidity also makes more motors available for binding, as expected, and as indicated by previous studies [35, 36, 45]. However, this effect, by itself, is significant only at lower ATP concentrations (and/or very high motor numbers) when the effective timescale for diffusion and binding is less than the unbinding time. In fact, we estimate that the reduction in interference is relatively the most important effect under physiological conditions of high ATP and low motor numbers. Taken altogether these effects can cooperatively result in increased processivity and collective mechanical force production against load with an increase in fluidity for teams of molecular motors, with both *in vivo* and technological implications.

## Results and discussion

### Surface fluidity reduces negative interference between bound kinesin motors

We first addressed the question of whether the fluidity of the cargo surface has any influence on the functioning of a bound kinesin motor. We expected that the motor attachment point's freedom to move on the lipid cargo surface in response to forces could suppress strain and reduce the forces experienced by the motor. We also expected that the surface fluidity would allow bound motors to alter their spatial distribution relative to the cargo and each other thereby also changing the forces they experienced. We started by looking at the positions of motor heads on the MT and anchor points on the cargo during a typical cargo run for a rigid and a lipid cargo (Fig 2a; see S1 Fig for a longer time window). From the motor head position subplots, it can be seen that motors on a rigid cargo are more spread apart and can be located opposite to each other relative to the cargo center of mass while on lipid cargos, bound motors tend to be closer and on the same side, in front of the cargo. This supports our expectation that the lipid membrane can alter the relative positioning of the motors which could lead to different strain forces on the motors.

To explore the forces experienced by motors more quantitatively, we computed the distribution of forces experienced by a single motor when it is part of a team of $n$ motors that are simultaneously bound to the microtubule. This distribution, shown for $n = 3$ bound motors in Fig 2b, is broader, reflecting larger forces, for a motor on a rigid cargo compared to any of the lipid cargoes. The lipid cargo motor force distributions is, in fact, comparable to that for a single bound motor carrying a rigid cargo. A rigid coupling of motors, therefore, results in individual bound motors experiencing higher forces than when they are singly bound to cargo, indicating negative interference. The introduction of surface fluidity appears to reduce this interference by effectively decoupling the motors, consistent with our expectations. This reduction is also apparent in the distribution of forces experienced both against (hindering) and in (assistive) the direction of processive motion (Fig 2c).

The reduction in forces due to fluidity is also evident in the fraction of non-negligible forces (S2 Fig), which we take to be forces with a magnitude $\geq 1\%$ of $F_s$, the stall force of the kinesin motor (S3 and S4 Figs show similar fractions in hindering and assistive directions). For rigid cargo, this fraction increases as a function of the number of bound motors, $n$, indicating negative interference between motors. In the presence of surface fluidity, however, no such increase in the fraction with $n$ is apparent, indicating a reduction of negative interference. In fact, the fraction decreases with $n$ at high surface fluidity ($D = 1 \ \mu m^2 \ s^{-1}$), which is likely due to the decrease in the average force experienced by the motors in the presence of multiple bound motors (S5 Fig). While the decrease in average force occurs for rigid cargo as well, the increase with $n$ due to negative interference dominates.

The variance between the individual forces experienced by simultaneously bound motors is also a measure of negative interference as it quantifies the deviation from perfect load-sharing where the variance should be zero. The mean variance over all sampled instances with $n$ bound motors shows a dramatic decrease at higher fluidity ($D = 1 \ \mu m^2 \ s^{-1}$) compared to the rigid cargo case (Fig 2d, see S5 Appendix for detailed description of this and other force related metrics used in this section). In fact, even the presence of a small amount of fluidity accounts for most of the significant drop in variance (S6 Fig) indicating that fairly good load-sharing can be achieved with modest fluidity. We also quantified negative interference more directly by studying whether individual motors were acting antagonistically, *i.e.*, exerting forces in opposite directions. To do this, we computed the correlation between the x-components of bound motor forces $\langle F_x^j F_x^k \rangle$ averaged over different motor pairs $(j, k)$ and time (Fig 2e). We see

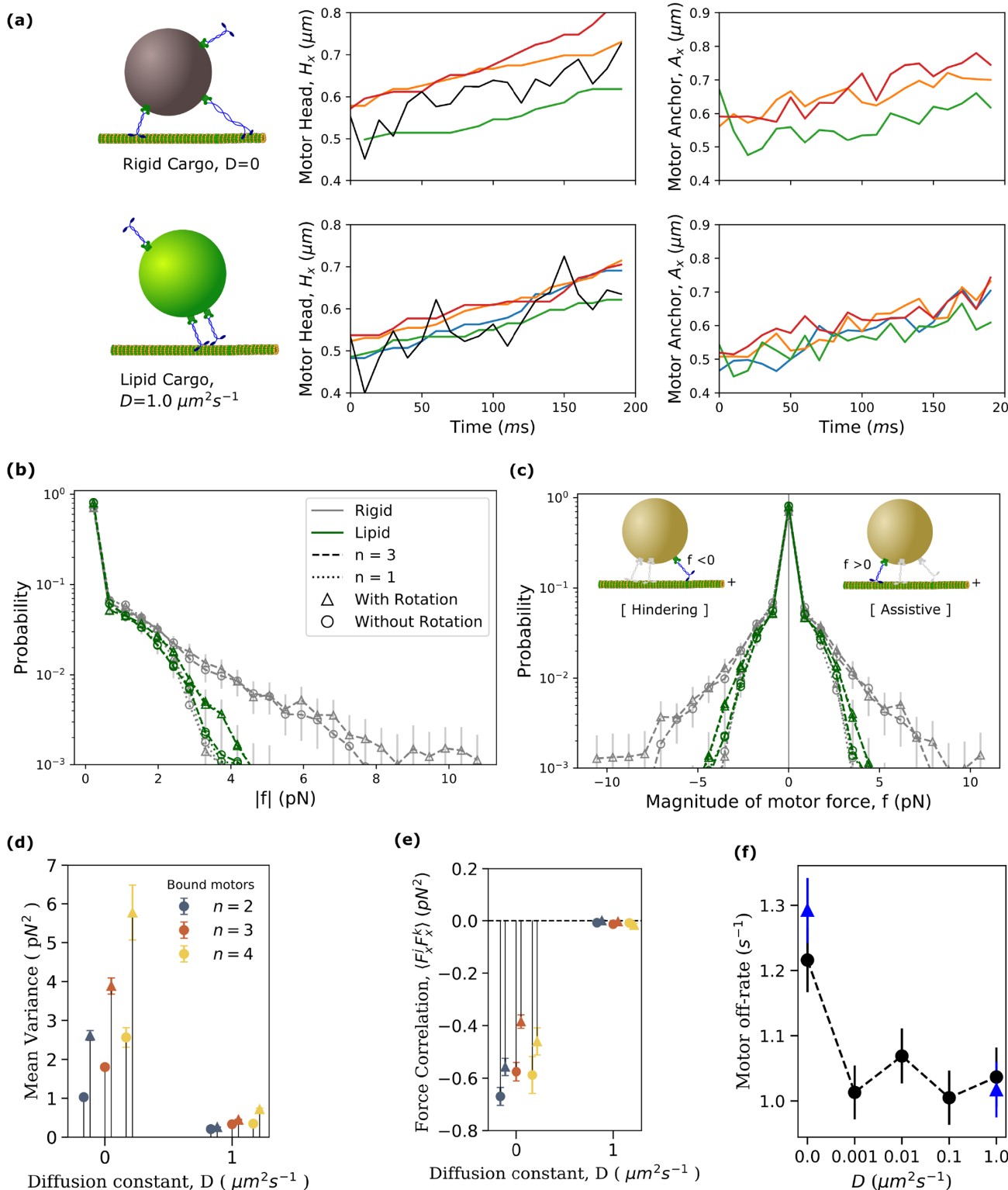

**Fig 2. Surface fluidity reduces the negative interference between bound kinesin motors which leads to reduction in motor off-rate. (a)** Typical trajectories for a rigid (top row) and a lipid cargo (second row). Motor head position subplots (middle column) include the x-positions of the center of mass of cargo (black lines) and head positions of motors on the microtubule (colored lines). The corresponding x-component of motor anchor positions (right column) on the cargo are also shown. **(b)** Distributions of the absolute magnitude of the force experienced by motors in rigid and lipid cargoes (including without cargo rotation) at a fixed number of bound motors ($n = 3$). Distribution for singly bound motor carrying a rigid cargo is also shown for

comparison. We first collect $S$ = 10000 force values randomly drawn from 200 cargo runs. We then bootstrap this (bootstrap sample size = 1,000 and the number of bootstrap samples = 10) to get the mean distribution and standard error of the mean. (**c**) Distribution of magnitude of motor forces when they specifically experience hindering loads ($f < 0$) and assistive loads ($f > 0$). Cartoon inside shows a possible scenario where the colored motor experiences hindering and assistive load. (**d**) The mean variance of forces among bound motors in rigid ($D = 0$) and lipid cargoes ($D = 1\ \mu m^2\ s^{-1}$) with (triangles) and without cargo rotation (circles). Mean variance was calculated as $\frac{1}{S}\sum_{i=1}^{S}\sigma_{|f|}^2(i)$ where the summation is over all the sample time points when the number of bound motors is $n$. $\sigma_{|f|}^2(i)$ is the variance in the magnitude of force experienced by the $n$ bound motors at $i^{th}$ sample. Sample size $S$ = 10000 drawn from 200 cargo runs. (**e**) Mean value of the correlation between the $x$-components of motor forces, $\langle F_x^j F_x^k \rangle$, averaged over motor pairs and time points with (triangles) and without cargo rotation (circles). Sample size $S$ = 10000 drawn from 200 cargo runs. (**f**) Average value of motor off-rate as a function of fluidity of cargo surface with (triangles) and without cargo rotation (circles). ($S$ = 580 from 200 cargo runs). Data for all the plots were obtained from the simulation of transport of cargoes with $N$ = 16 motors at high ATP concentration of 2 mM. Data sampling rate was 100 $s^{-1}$.

that this correlation is highly negative for rigid cargo indicating opposing motor forces while it is negligible in the presence of fluidity indicating motors on the same side of the cargo pulling in the same direction. Taken together, our results show that interference dominates over load sharing for rigid cargo while increasing surface fluidity reduces this interference and promotes load-sharing.

Next, we addressed the implications of the reduction in negative interference for the functioning of motors in the context of transport. The off-rate of a kinesin motor increases with the magnitude of the force that it experiences and also depends on its direction [15, 23, 46–48]. The observation that surface fluidity reduces negative interference leads us to expect that this will also lead to lower off-rates for individual motors. We measured the mean off-rate of motors as the inverse of the mean time spent by individual motors between a binding and subsequent unbinding event. We found that bound motors on fluid cargoes do indeed have lower off-rates than those on rigid cargoes (Fig 2f). Consistent with our findings for the forces and interference, the introduction of a small amount of fluidity accounts for most of the significant drop in off-rates with no significant variation with increasing diffusivity.

The forces that motors feel and the off-rate of motor is a function of the spring constant of the kinesin coiled-coil region. In fact, previous studies [28, 49] have found that off-rate of ensemble of motors and the transport properties of cargo are sensitive to the stiffness of the motor. Our simulations (see S22 Fig) indicate that average single motor off-rate can increase by almost 20% for a change in the motor stiffness from 0.1 pN/nm to 0.5 pN/nm for rigid cargo, while for lipid cargo the change is not significant, which is again consistent with the picture of the lipid membrane suppressing strain in the motors.

We also ran simulations at different cargo sizes to determine how the single motor metrics change. The distribution of forces (S7(a) and S7(b) Fig) shows no clear trend except perhaps for a marginal increase in larger forces with decreasing radii for lipid cargo, which could be due to the steric constraints becoming more important. Interference between motors, quantified by the force correlations (S7(c) Fig), show negligible correlations between motors for lipid cargo at any size, while for rigid cargo, the interference appears to increase with size, consistent with potentially increased spacing between motors. Motor off-rate (S7d Fig) decreases as a function of cargo size for both rigid and lipid cargo but the off-rate of motor in lipid cargo is lower than rigid cargo for all cargo sizes, which is again consistent with picture of reduced interference and greater load-sharing with a lipid cargo.

We also took the opportunity to explore the impact of cargo rotations on motor forces. While cargoes do rotate considerably during the course of a typical cargo run and especially at intersections or obstacles [50, 51], it is not clear whether this rotation impacts transport metrics like motor off-rate, force distribution and cargo runlength. Answering this question is also interesting from a modeling perspective and helpful for improved analytical approximations. Disallowing rotational motion of the cargo in our simulations leads to a slight decrease in the

magnitude of motor forces, as is apparent in $n$ = 3 cases shown in Fig 2b and 2c. In our simulations and in *in vitro* experiments in aqueous media, the load due to viscous forces is low and the origin of higher forces is thermal fluctuations [52] and negative interference between motors. The higher forces experienced by motors in the presence of rotation is therefore likely due to rotational diffusion adding additional noise into the cargo dynamics. It is to be noted that the reduction in forces due to removing cargo rotations is slightly more pronounced for rigid cargo but in both cases the reduction is small compared to the reduction due to fluidity. The variance of motor forces (Fig 2d) also decreases when we remove rotational diffusion because of the decrease in the noise though the effect is not significant for lipid cargo. Similarly, the magnitude of force correlations (Fig 2e) increased when we removed rotation for rigid cargoes while the correlations are essentially the same and close to zero for lipid cargo. These effects arise because the removal of rotational diffusion removes a source of uncorrelated noise. Finally, we compared the change in motor off-rate with and without incorporating rotation in our model (Fig 2f). We notice that rotation may increase the off-rate slightly (by about 5%) for rigid cargo, while the off-rate of motors on lipid cargo is unaffected by cargo rotation. Thus, it seems that ignoring rotation of cargo leads to no qualitative changes and is a perfectly good quantitative approximation for lipid cargo and a reasonable one that slightly underestimates forces and off-rates for motors on a rigid cargo. It is to be noted that rotations can play a much more important role in enhancing binding rates from solution as seen in [36], but for lipid cargo, the diffusion of motors dominates over rotations of the cargo, and therefore rotations play a negligible role. In what follows, results presented include rotation of the cargo and are essentially very similar to the results without rotation unless explicitly noted.

Overall, our modeling indicates that increasing surface fluidity reduces inter-motor interference and promotes load-sharing leading to reduced off-rates and potentially longer processive runs for individual motors.

## On-rate of a motor increases with the number of bound motors for fluid cargo

The run length of a cargo carried by a team of motors depends on the interplay between the off-rates and on-rates of individual motors [23]. The on-rate of a motor, typically considered to be independent of the number of other bound motors, depends on its intrinsic binding rate when it is within reach of the MT. The region of the cargo surface (or access area) from where an anchored kinesin can reach the MT is defined, in our model, by the set of points on the surface at a distance less than or equal to the rest length of the motor ($L_{mot}$ = 57 nm). For a cargo bound to the MT with one or more motors, this access area is a function of the cargo geometry and the proximity of the cargo to the MT [35, 38, 53]. For rigid cargo, motors can only bind if they happen to be anchored in the access area while for fluid cargo, motors can diffuse in and out of the access area. For a given motor type with a fixed intrinsic binding rate and a given cargo geometry (a sphere here), the access area is then simply a function of the distance of the cargo from the MT. Interestingly, our measurements of the average distance of the center of mass of the cargo from the MT indicated that the cargo comes closer to the microtubule as the number of bound motors increases (Fig 3a). This decrease in distance arises from the increase in the net vertical component of the average force due to an increased number of motors which serves to counteract cargo fluctuations away from the MT. We also note that cargoes with low surface fluidity are closer to the MT than the cargoes with high surface fluidity, especially for high numbers of bound motors. This is because motors on fluid cargoes can relax their tension by sliding on the cargo surface allowing for larger fluctuations (S8 and S9 Figs) and therefore a higher average distance from the MT. We also note that rotations of the cargo

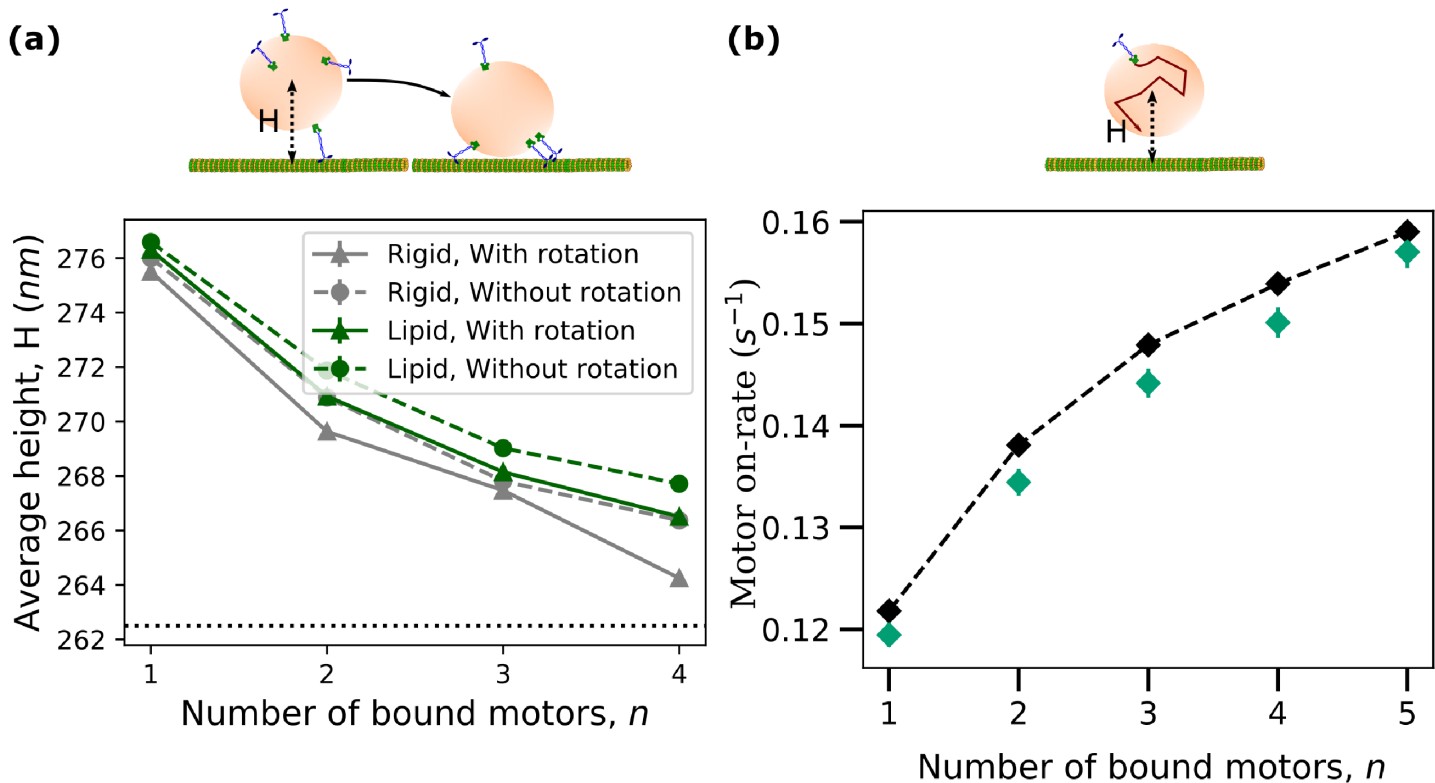

**Fig 3. On-rate of a motor increases with the number of bound motors for fluid cargo. (a)** Average measured distance (H) of the center of mass of cargo from MT as a function of the number of bound motors and fluidity. Horizontal line indicates the height below which the cargo experiences steric interaction from the MT. Data were obtained from the simulation of transport of cargoes with $N = 16$ motors at high ATP concentration of 2 mM. Data sampling rate was $100\ s^{-1}$. **(b)** Computational measurements of single motor on-rate (green diamonds) and analytical approximation (black dashed line with diamond), ($D = 1\ \mu m^2\ s^{-1}$, $S = 10000$) using the cargo height for a given $n$ from the data in (a). Error bars for all plots represent the standard error of the mean. Rotation of cargo was not considered in the on-rate measurement simulations for (b).

seem to allow the cargo to get slightly closer to the MT, presumably because the extra forces due to rotational diffusion serve to increase the net vertical component of the average force as well.

Thus, for both rigid and fluid cargo, as the number of bound motors increases, the cargo approaches the MT leading to an increase in the access area. We expect that this should lead to an increase in the on-rate of an individual unbound motor because its likelihood of being in the access area is increased. We verified this by measuring the on-rate of a motor diffusing on the surface of a cargo held at specific distances from the MT (Fig 3b) that correspond to different numbers of bound motors (from data in Fig 3a for $D = 1\ \mu m^2\ s^{-1}$). Here we chose a high diffusion constant for simplicity and measured the mean time required for a randomly placed motor to diffuse and bind to the MT. Indeed, we see that the effective on-rate of the motor, calculated as the inverse of this mean time, increases with decreasing distance corresponding to increasing numbers of bound motors (Fig 3b). To show that the quantitative increase in on-rate was explained by the decreasing distance, we also analytically estimated the change in on-rate. Since the average time spent by a motor in the access area before binding to the microtubule is approximately proportional to the ratio of access area to total area in the high diffusion constant limit, we estimated that the effective on-rate is $\pi_{ad} = \pi_0(S_a/S_T)$, where $\pi_0 = 5\ s^{-1}$ is the intrinsic binding rate, $S_a$ is the access area and $S_T = 4\pi R^2$ is the total surface area of the cargo. Such an assumption is reasonable for cargo surfaces with high diffusion constants where the

time scale for motor diffusion across the cargo surface is small as seen in previous studies [35, 38, 54]. By computing the access area numerically for different cargo distances, we were able to estimate $\pi_{ad}$ corresponding to different numbers of bound motors. The rather good agreement with the measured values from simulations (Fig 3b) indicates that the increase in on-rate is captured by the effects of increased access area. It is to be noted that the rate at which the cargo transitions from being bound by $n$ to $n + 1$ motors can be calculated from this on-rate, the off-rate and the number of unbound motors available for binding (details in S2 Appendix and data in S10 Fig).

Our results reveal a positive feedback effect at play. As the number of bound motors increases, the cargo gets closer to the MT and the on-rates of each of the remaining motors increase which further increases the number of bound motors. Interestingly, the apparently small change in cargo height can have a big effect on the overall processivity of the team as discussed in the subsequent section on the contributions of different effects to overall run length.

However, it should be noted that this positive cooperativity that we observe is conditioned on our assumption that an unbound motor can bind to any position on the microtubule at a distance less than the rest length of the motor ($L_{mot}$). For situations where a large number of protofilament tracks are not available, such as myosin motors on actin filaments, steric occlusion could be important. For instance, studies [54] on myosin binding to actin also found that the cargo comes closer to the filament with an increase in the number of bound motors, but this leads to a decrease in the on-rate of motors because of increased steric occlusion from the cargo itself.

## Surface fluidity increases the availability of motors for binding

A cargo cannot, however, benefit from the positive feedback noted in the last section, unless it makes a sufficient number of motors available for binding. We expect that the diffusion of motors on the cargo surface should lead to a greater availability of motors in the access region and hence a higher number of bound motors compared to rigid cargo with the same overall number of motors. We also expect that this will be true only if $\tau_{bind}$—the typical time for any one of the unbound motors to bind to the microtubule is less than $\tau_{off} = 1/\epsilon_0$—the typical time for a given bound motor to detach. The relative values of these two timescales are set by parameters such as the number of motors ($N$), the radius of cargo ($R$), and the ATP concentration. For instance, raising the motor density (say by raising $N$) will result in a higher effective on-rate and a decreased $\tau_{bind}$, while decreasing the ATP concentration, lowers the off-rate and raises $\tau_{off}$.

We first looked at a parameter set ($N = 16$ [ATP] = 2 mM) satisfying the condition $\tau_{bind} < \tau_{off}$ (see S2 Appendix for details of estimates). Fig 4a shows that, while the average number of bound motors, $\bar{n}$, increases with time for both rigid and fluid cargoes, the fluid cargo indeed accumulates more motors. The spatial distribution of motors in Fig 4b further highlights the difference between rigid and fluid cargoes with the probability of finding motors near the microtubule on a fluid cargo being larger at late times than for rigid cargo. It is to be noted that the slight increase at late times for rigid cargoes is simply due to the higher contribution to the average of cargoes, that happen to have motors clustered together, surviving for longer times.

To further test the dependence of motor accumulation on the relative values of the two time scales, we considered four other sets of physiologically relevant parameters of $N$ and [ATP] that had different relative magnitudes of $\tau_{off}$ and $\tau_{bind}$ (shown in Table A in S2 Appendix) As predicted, there is no statistically significant difference in the average number of bound motors as a function of diffusivity (see Fig 5a) for cases when $\tau_{off} \approx \tau_{bind}$ ($N = 4$,

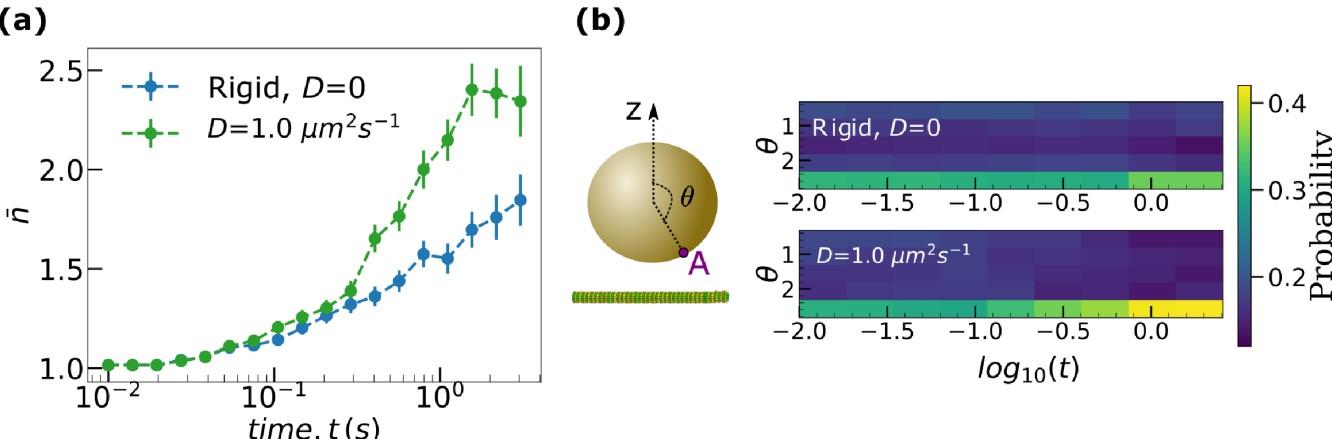

**Fig 4. Surface fluidity increases the availability of motors for binding. (a)** Ensemble average of the number of bound motors, $\bar{n}$ as a function of time (s). Error bars represent the standard error of the mean. **(b)** Spatial distribution of the kinesin motors on cargo, computed as the probability of locating motors as a function of the polar angle $\theta$ and time, $t$. $\theta$ is the polar angle of the motor, measured with respect to $z$ axis perpendicular to the microtubule and passing through the center of the cargo. Data was obtained from the transport of cargoes with a total of $N = 16$ motors at [ATP] = 2 mM.

[ATP] = 2 $m$M and 100 $\mu$M). The number of bound motors, however, increases as a function of diffusion constant ($D$) for cases where $\tau_{bind} < \tau_{off}$ ($N = 16$ [ATP] = 2 $m$M and 100 $\mu$M, $N = 4$ [ATP] = 4.9 $\mu$M) as expected. This supports our expectation that fluid cargoes can indeed accumulate higher number of bound motors than rigid cargoes, but only under conditions that ensure that $\tau_{bind} < \tau_{off}$, i.e. when the number of motors ($N$) is high or the ATP concentration is low or both.

## Fluid cargo have longer run lengths than rigid cargo

So far we have shown that increasing surface fluidity reduces inter-motor interference and promotes load-sharing leading to reduced off-rates. Fluidity also makes more motors available for binding in a cooperative fashion with increased binding leading to higher on-rates. We would expect that having more motors bind that survive longer should lead to increased run lengths for fluid cargo. The run lengths of cargoes measured as a function of diffusion constants (Fig 5b) suggest that this is indeed possible given the right ATP concentration and the number of motors on the cargo (see S13 Fig for additional diffusion constants). Although the run length is unaffected by fluidity when $\tau_{bind}$ is of the order of $\tau_{off}$ (low $N$ and medium/high [ATP]), the run length increases with the fluidity of the cargo surface when $\tau_{off} > \tau_{bind}$ (for high $N$ or low [ATP] or both). The effect is much more pronounced when the disparity between the timescales is higher.

We saw earlier that the average number of bound motors shares similar trends (Fig 5a) as a function of cargo surface fluidity. It is to be noted, however, that cargoes can have the same average number of bound motors but different run lengths. For example, although high fluidity ($D = 1\,\mu m^2\,s^{-1}$) cargoes with $N = 4$ at [ATP] = 4.9 $\mu$M and $N = 16$ at [ATP] = 2 mM have the same average number of bound motors, the lower ATP case with fewer total motors shows a substantially larger run length (Fig 5a and 5b). This is a consequence of the higher tendency of a cargo to survive a 1 motor state at low ATP due to a lower motor detachment rate (more details in S4 Appendix).

For saturating ATP conditions, we note that the observation that a high number of motors (16 here) achieves a runlength of only about 2 $\mu m$ is consistent with values predicted in other *in vitro* experiments and models [35]. This is because even though there are 16 motors on the

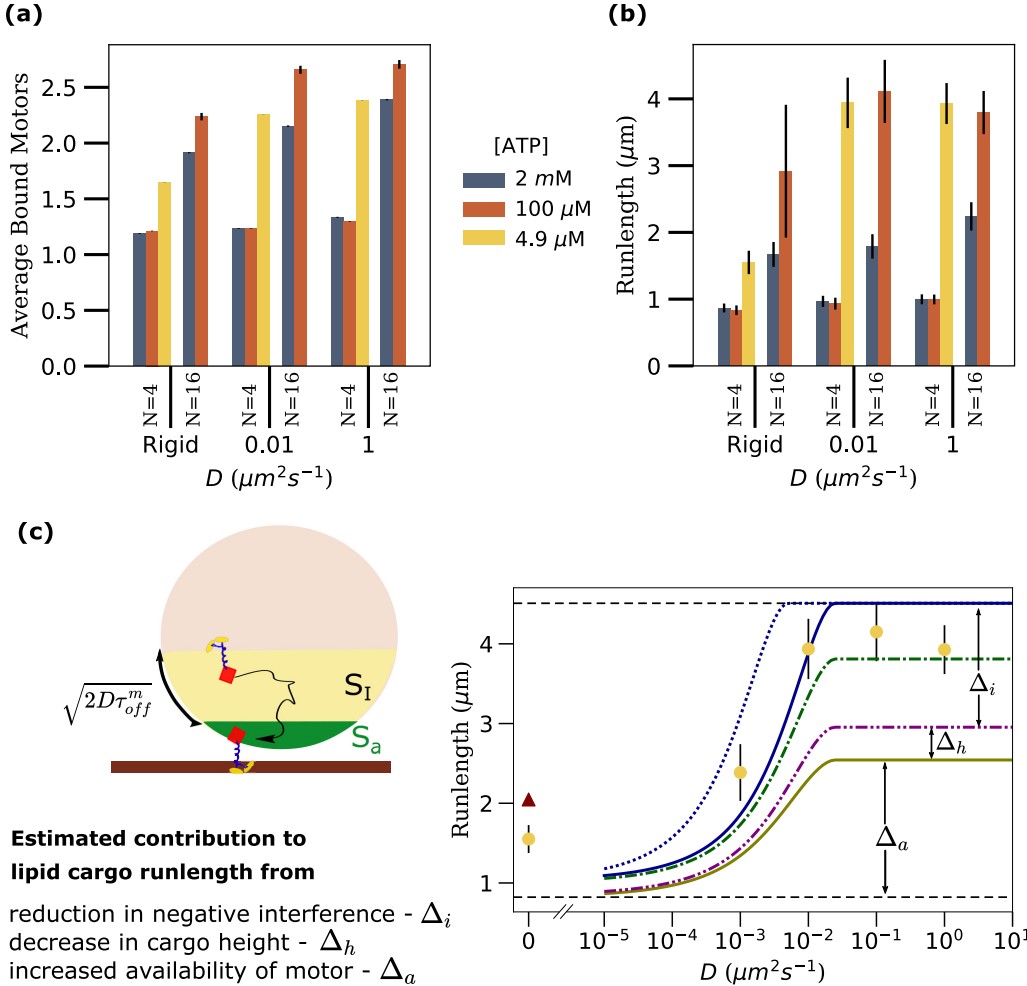

**Fig 5. Fluid cargo have longer run lengths than rigid cargo. (a)** Average number bound motors, **(b)** Runlength as a function of diffusivity ($D$) for two different numbers of motors on the cargo ($N$) and three different ATP concentrations. 200 cargo runs were considered for each parameter set. Error bars represent the standard error of the mean (SEM). The average number of bound motors in (a) was calculated as the mean of the number of bound motors ($n$) in all time samples (with data sampling rate = 100 s⁻¹) of these 200 cargo runs. Rotational diffusion of cargo was not considered in these simulations. The sample size was large and hence the error bars obtained as the standard error of the mean is very small. **(c)** Comparison between run lengths from our simulations ($N = 4$, [ATP] = 4.9 $\mu$M, yellow circles) and analytical estimates (maroon triangle, solid, dashed and dash-dotted lines) as described in the text. *Left cartoon*: A lipid cargo, with access area ($S_a$, green) and influx area ($S_I$, yellow) shown. Influx area is defined as the region within $\sqrt{2D\tau_{off}^m}$ from the access area.

cargo, the average number of motors that can access the microtubule is only about 1.37 ($N_a = 1 + (S_a/S_T)$). Our runlength values are also comparable to simulation results by Bovyn *et. al.* [36] for rigid and lipid cargoes under saturating ATP conditions. Finally, the range of runlength values for different numbers of motors for a cargo of radius 250 nm agrees with the range of values from experimental studies [44], lending further validity to our model.

While the results in Fig 5 are from simulations without considering cargo rotation, we also ran simulations with rotation and compared the runlength and average number of motors between these two cases S20 Fig. There is no noticeable change in these metrics with and without cargo rotation.

Overall, our results indicate that surface fluidity increases run lengths when the number of motors ($N$) is high or the ATP concentration is low or both. While previous studies have shown that the runlength of multi-kinesin rigid cargoes increases with decreasing ATP concentration [55], our simulations show explicitly that the increase in runlength with decreasing ATP concentration is significantly larger for lipid cargoes than for rigid cargoes.

## Contributions of different effects to overall run length

We now consider the relative contribution to the overall run length of the different effects we showed are at play; the reduction in negative interference, the cooperative increase in on-rates and the increased availability of motors.

**Analytical expression for run length.**   To estimate the magnitude of these contributions, we first consider an analytical expression for the run length [23]

$$r = \frac{v_o}{N_a \pi_{ad}} \left[ \left( 1 + \frac{\pi_{ad}}{\epsilon_m} \right)^{N_a} - 1 \right] \tag{1}$$

where $v_o$, $N_a$, $\pi_{ad}$, $\epsilon_m$ are the motor speed, the available number of motors for transport, the single motor binding rate and the mean motor off-rate respectively. Based on our results, we now generalize the expression by incorporating the dependence of these parameters on the cargo surface diffusivity $D$ and the number of bound motors.

The mean off-rate, $\epsilon_m$, is sensitive to the presence of inter-motor interference and the introduction of fluidity almost completely eliminates interference (Fig 2b–2f). This means we can treat $\epsilon_m$ as having two values; a higher one for rigid cargo due to interference and a lower one for fluid cargo (Fig 2f).

The number of available motors, $N_a$, is usually taken to be the number of motors in the access area, $S_a$, for rigid cargo. For $D > 0$, however, additional motors from an influx area $S_I$ (see Fig 5c) can reach the access area before the cargo unbinds from the microtubule and contribute to transport. Here we estimate $S_I$ as defined by a region within a distance $\sqrt{2D\tau_{off}^m}$ of the access area, representing the distance diffused by a motor over the mean motor lifetime $\tau_{off}^m = 1/\epsilon_m$. We, therefore, take $N_a$ to be the average number of motors in the access and influx areas combined, $N_a = 1 + (N-1)(S_I + S_a)/S_T$. Thus the number of available motors starts from a minimum value $N_a = 1 + (N-1)(S_a/S_T)$ at $D = 0$ and increases monotonically as function of $D$ before saturating at $N_a = N$.

Finally, motors bind with a rate $\pi_{ad} = \pi_0 = 5\ s^{-1}$ when they are within the access area and do not bind otherwise. This means we can take the mean on-rate of the available motors to be $\pi_{ad} = \pi_0 S_a/(S_I + S_a)$. We note that the access area, $S_a$, depends on cargo geometry and in particular on the height of the cargo above the microtubule. As more motors bind, the height decreases, $S_a$ increases and the effective on-rate increases (Fig 3). Finally, we note that the cargo velocity did not change by more than 5% over the range of parameters we tested (S15 Fig) and so we took $v_o$ to be constant.

**Approximate bounds for run length.**   Using these analytic estimates, we can compute approximate bounds for the relative contributions of the different effects to overall run length. As an example, we do this for the case of low motor density and low ATP ($N = 4$ [ATP] = 4.9 $\mu$M, Fig 5c, other cases shown in S17 Fig). We start with the limiting lower bound case where there is no additional availability of motors due to diffusion ($N_a = 1 + (N-1)(S_a/S_T)$), no reduction in interference ($\epsilon_m = 0.12\ s^{-1}$, S16 Fig) and the cargo height is at its maximum (corresponding to having 1 motor bound (Fig 3a)). This yields the horizontal dashed line at the bottom in Fig 5c. A similar upper bound that assumes all motors are available ($N_a = N$), interference is absent ($\epsilon_m = 0.1\ s^{-1}$, see S16 Fig) and the cargo height corresponds to that at $n = 2$

(which is close to the average number of bound motors for this configuration), yields the horizontal dashed line at the top. While the bounds bracket the run lengths measured from the simulations, we notice that the lower limit is well below the actual values, especially at higher diffusion constants. It is to be noted that the bounds do not really extrapolate well to $D = 0$ because, for rigid cargo, the average run length is not set by the average number of motors in access area. A better approximation is the weighted average of run lengths for different numbers of motors in the access area weighted by the probability of having that many motors in the access area (maroon triangle in Fig 5c; details in S3 Appendix).

**Comparison of contributions from different effects.**   We first consider the contribution due to diffusion increasing the availability of motors alone. Allowing $N_a$ to increase with $D$ but with no reduction in interference ($\epsilon_m = 0.12\ s^{-1}$) and the maximum cargo height, corresponding to $n = 1$, yields the lowest dash-dotted curve in Fig 5c. While there is a substantial increase in run length with increasing $D$, this increase (of order $\Delta_a$ at high $D$) yields an estimate that is still significantly below the measured values at high $D$. This indicates that the increased availability of motors alone cannot account for the entire increase in run length. We next incorporate the cooperative increase in the accessible area due to more motors binding and the cargo moving closer to the microtubule by setting the cargo height to that corresponding to $n = 2$ bound motors. This yields an increase in the run length (middle dash-dotted curve in Fig 5c) but the gain, of order $\Delta_h$ at high $D$, is also too small to account for the observed increase. We finally incorporate the effect of reduced interference between the motors due to surface fluidity by taking $\epsilon_m = 0.1\ s^{-1}$. This shifts the run lengths upwards considerably, by $\Delta_i$ at high $D$, and brings the estimates to within the range of the observed values (solid curve in Fig 5c; the upper dash-dotted curve is for a cargo height corresponding to one bound motor). Intriguingly, the surface fluidity based reduction in negative interference has an effect that is comparable or even stronger (see S17 Fig also) than the effect due to the increase in the number of available motors. We note that a reduction in the interference and the resultant reduction in mean off-rate leads to an increase in the mean motor lifetime, $\tau_m$, and hence an increase in the influx area $S_I$ which is set by $\sqrt{2D\tau_{off}^m}$. Thus, reduction in interference contributes to an increased run length not only due to an increased lifetime of each motor but also due to the resultant increase in the number of motors available for transport. While we used the single motor lifetime $\tau_{off}^m$ to estimate $S_I$, this represents a lower bound since new motors may reach the access area over the entire cargo runtime $\tau^c$. Using the saturating value of $\tau^c$ (for high $D$) in place of $\tau_{off}^m$ in the expression for $S_I$ therefore gives an upper-bound estimate for the run length (upper dotted curve in Fig 5c). We note that the top three curves for estimates that include the effects of reduced interference and increased availability of motors reflect the measured values of run length fairly well as a function of increasing surface diffusion constant across different conditions of ATP and motor number (see S17 Fig).

## Motor teams under external load

So far, we have considered the only load on the motor teams to come from viscous drag on the cargo which is very low for the aqueous medium we have been considering and usually dominated by thermal fluctuations. We now consider the behavior of the motor teams under external loads, specifically looking at motor clustering and collective force generation.

**Motors dynamically cluster at specific locations on the lipid cargoes.**   Past studies of force generation by motors coupled to lipid membranes have shown that the motor performance (as measured by the velocity of microtubules) is reduced due to "slippage" of motors on the lipid membrane [40]. Also, studies have shown that rigid-like ordered regions on the membrane are beneficial for transport of cargo because the motors can anchor on these ordered

regions and use this as a rigid substrate to effectively generate forces [37, 38]. Hence, although motors on lipid cargo can interfere less and survive longer, a question arises as to whether the motors can efficiently generate forces against load while hauling cargo.

An important thing to note is that the geometry of the lipid membrane is a spherical surface, or at least a closed, bounded surface, for cargo *in vivo* such as vesicles and organelles. For such cargo, there is a limit to how far the motors can slip. Motors can slip only as long as the tangential force is non-zero. Once the tangential component reduces to zero and all the force is radial, a motor on a lipid cargo should be as efficient in generating force as on any other substrate.

To understand whether motors are likely to be in this radial force regime, we looked at the probability distribution of the location of motors on the cargo surface. To obtain this distribution, we ran simulations of cargo transport (N = 16, High [ATP] = 2mM) under different hindering loads on the cargo and recorded (sampling rate of $100 \ s^{-1}$) the anchor positions of MT bound motors relative to the body coordinate axis of the cargo (see schematic in Fig 6). Heat maps in Fig 6 show the probability distributions of the anchor positions of MT bound motors on the cargo surface. We can clearly see that there is a preferred position for motors on the cargo surface at different hindering loads (Fig 6). At zero hindering load, motors experience only the viscous load which is small enough that this distribution is broad and only a diffuse enhancement is visible near the MT (bottom of the cargo) However, as we increase the cargo load, motors increasingly appear at a preferred location on the cargo surface to balance the load. We calculated the location on the cargo where the tangential force reduces to zero and the horizontal component of force balances the hindering load and found that the height (z-coordinate) of this location above the MT increases with increasing hindering load (see S5 Appendix). The range of locations we obtained from this argument were in reasonable agreement with the values we measured from simulations (see S5 Appendix for calculation and values) validating the mechanism for preferential localization of motors. While we see motor density enrichment for both rigid and lipid cargoes with increasing hindering load, we note that the lipid membrane can allow motors to slide to this preferred location while this is not possible in rigid cargo, explaining the higher density enhancement in the lipid case. Motors can therefore dynamically cluster at a preferred location on the lipid cargo which minimizes the tangential component of motor forces and leads to better load-sharing. Our results are consistent with prior observations of dynein clustering in axonemal endosomes and related modeling [56]. This sort of dynamic clustering can exist in addition to any intrinsic clustering mechanisms that may exist for *in vivo* lipid membranes [36–38] which can enhance processivity in motor teams.

This dynamic clustering of motors may also lead to the deformation of the vesicle. Collective forces due to kinesin motors have been observed to lead to the extraction of membrane tubes out of vesicles [43, 57]. There is a threshold of point force on membrane below which such a membrane extraction is not possible (see S1 Appendix). When the hindering load is low, motors are not localized, the force is low and not concentrated and the threshold is not reached, which is the reason we can ignore deformation of vesicles under normal cargo transport conditions (see S1 Appendix). If the hindering load exceeds the threshold, the clustering of motors may lead to membrane deformation and tube extraction. It should be noted that in our model, motors can come arbitrarily close to each other, while in reality, volume exclusion could broaden the localization region slightly, thereby affecting the threshold for tube extraction.

**Motor teams in fluid cargoes generate higher collective forces.** We have predicted that lipid cargoes may have a higher number of MT bound motors which exhibit less negative interference, better load sharing and increased processivity. We have also predicted that

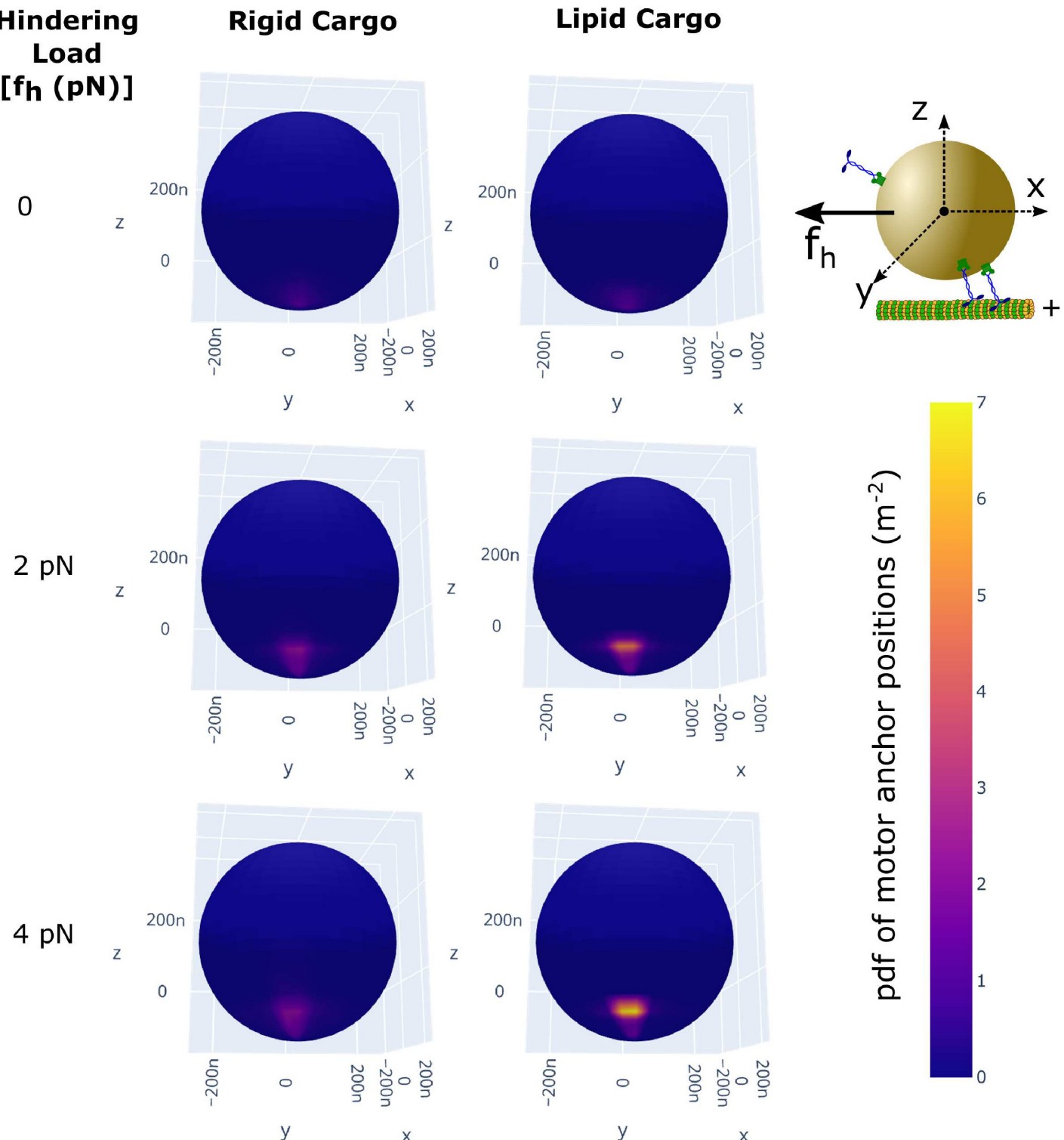

**Fig 6. Motors dynamically cluster at specific locations on the lipid cargoes.** Average distribution of bound motors on the cargo surface at different hindering loads on the cargo ($f_h$). Only multiple bound motor cases were considered. Data is from 200 cargo runs with N = 16, high [ATP] = 2 mM with anchor positions of bound motors recorded at a sampling rate of 100 $s^{-1}$. Anchor positions are measured relative to the body coordinate axis of the cargo (schematic at top-right). Diffusion constant of motors on lipid cargoes was 1 $\mu m^2 \, s^{-1}$.

motors can cluster together on lipid cargo under constant increased hindering load. Do all these factors imply that lipid membrane coupled motors can also generate more collective force?

Collective forces generated by teams on cargo have been measured by optical trap experiments [58] for motor-teams *in vitro* as well as *in vivo*. *In vitro* experiments have indicated that the kinesin-motor teams generate sub-additive forces [30, 59, 60] whereas *in vivo* experiments [60, 61] have shown that motors exert additive forces. While this difference in cooperativity might arise from differences *in vitro* and *in vivo* environments, we wanted to check if it could arise from the difference in the cargo surface properties in *in vitro* (rigid cargoes) and *in vivo* experiments (lipid cargoes).

To this end, we modeled cargo placement in an optical trap of force constant (0.06 pN/nm) [61] (see cartoon in Fig 7a) and ran simulations of cargo transport starting from a single microtubule bound motor for a maximum of 10 s (or until all motors unbind) for a total of 200 cargoes. We chose the parameters $N = 4$, $[ATP] = 2$ mM, which we believe is accessible to experiments [44]. We measured the values of the average total x-component of force due to motors on the cargo. We then rescaled the collective force [60] by $N \times F_s$ to get the rescaled average force per motor. The rescaled average force tells us how well the individual forces add up and whether or not the motors detach before reaching their stall force, serving as a good measure of the cooperativity between motors.

We scanned through the parameter space of single motor unloaded velocity ($v_o$) and stall force ($F_s$) (see Fig 7b) to get a clear picture of motor cooperativity in different parameter regimes and to allow for comparisons with previous studies [60]. For both rigid and lipid cargoes, we observe that the rescaled average force is higher for lower stall forces and higher velocity in agreement with previous coarse-grained computation for multi-motor teams [60]. However, we find that motors on lipid cargo seem to generate higher rescaled average forces compared to those on rigid cargo virtually over the entire phase space explored (see Fig 7c). Interestingly, the rescaled average force that we measure in the lipid cargo case is comparable to previous estimates that assumes perfect load sharing [60]. This is again an indicator of improved load sharing properties in lipid cargoes.

Thus the observations of sub-additive forces in *in vitro* experiments [30, 59] and additive forces in *in vivo* [61] experiments could be due to differences in the fluidity of the cargo surface. More controlled experiments on membrane-bound and membrane-free cargoes [44] could verify our prediction.

## Conclusion

Our results show that motors coupled through a rigid cargo experience a broader range of forces when multiple motors are bound, reflecting the geometric constraints on the motor-cargo attachment points leading to antagonistic forces and therefore negative mechanical interference between motors. This interference is consistent with literature findings [28–30] and counteracts the expected load-sharing with increasing numbers of motors. We predict that the increased fluidity of the lipid membrane which allows the attachment points to move, suppresses strain, decreases the negative interference between kinesins and permits load-sharing between motors to be more effective. The reduction of negative interference not only contributes almost equally to run length at low ATP (and/or high motor numbers) as increased motor availability but is, in fact, the dominant effect at saturating ATP and low motor numbers (see S17(a) Fig) which is typically the normal physiological (and *in vitro* experimental) state. Direct measurements of this enhanced load-sharing may be possible using optical trap experiments with an applied load on rigid and fluid cargo. Tuning the load and motor number to a

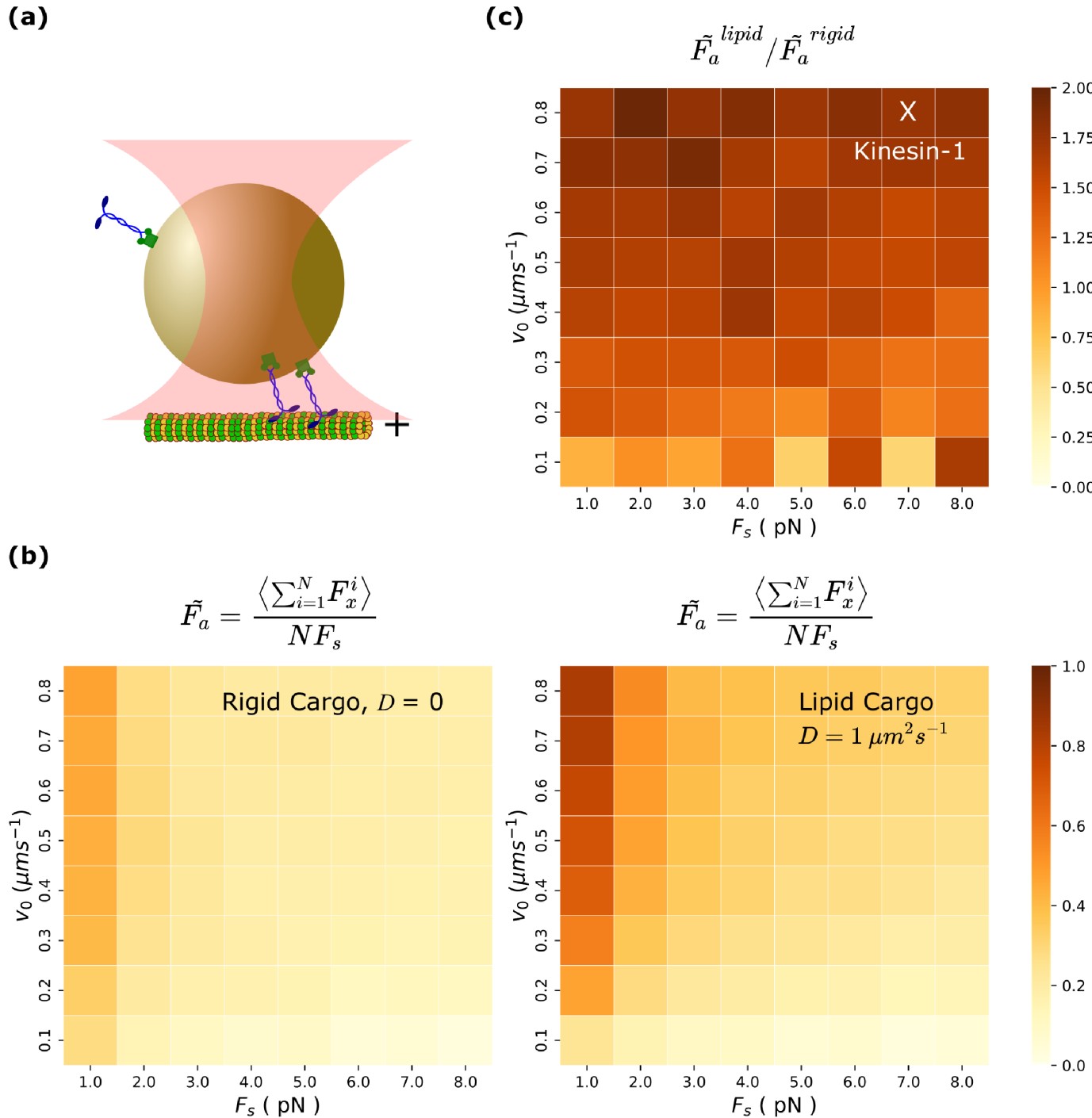

**Fig 7. Motor teams in fluid cargoes generate higher collective forces. (a)** Schematic of the optical trap set up that we used in simulation. **(b)** Rescaled average collective force $\tilde{F}_a = \frac{\left\langle \sum_{i=1}^{N} F_x^i \right\rangle}{NF_s}$ as a function of unloaded motor velocity ($v_0$) and stall force of motor ($F_s$) in rigid (left) and lipid (right) cargo. **(c)** Ratio of $\tilde{F}_a$ between lipid and rigid cargo. 200 cargo runs were used for each parameter set. For each cargo run, data of motor forces for each microtubule bound motors were recorded at a sampling rate of 100/s for a maximum time of 10 s for each cargo run. Average collective force is calculated as the average of the total x-component of motor forces over all time samples and cargo runs and then rescaled by $NF_s$. N = 4 and [ATP] = 2 mM.

regime where enhanced load-sharing can decrease motor off-rates may be able to resolve the effects in comparisons of rigid versus fluid cargo.

The effects of inter-motor interference in rigid cargo are also expected to be seen in the measurements of speed. However, while our measurements of speeds from the simulations do show a decrease with increasing numbers of bound motors (S15 Fig), it is not as dramatic as reported in [44]. Consistent with [44], we find this decrease to be negated in the presence of fluidity but the effect is not statistically significant. One possibility is that the force-velocity relation we used (Eq 17 in Materials and Methods) is not sensitive enough at low loads. Another more intriguing possibility is that the experimentally observed speed-up of lipid cargo [44, 45] may be due to a dispersion of speeds among motors leading to a bias for faster motors in the lead, and slower motors trailing and unbinding more often. On the other hand, microtubule-assay experiments [40] have indicated that the gliding velcoity of microtubules is decreased when motors were anchored on a membrane because the motors "slip" on the membrane. This seems to contradict velocity measurements from both the membrane-bound cargo transport experiments [44, 45] and our simulation. However, this disagreement can be attributed to the difference in the geometry of lipid bilayer between assay set-up [40] and our cargo transport set-up. In a spherical geometry, like our cargo, motors slip only until the tangential component of force goes to zero, while, on a flat membrane, a motor slips every time it makes a step.

Since the reduction in interference manifests as a decrease in the off-rate of motors, it leads to increasing numbers of bound motors at higher membrane diffusion constants. Interestingly, we found that the on-rate was not constant, but increased with an increasing number of bound motors that effectively pulled the cargo surface closer to the microtubule. Monitoring the binding times of successive motors in an optical trap geometry could potentially allow for the experimental verification of this effect. This interesting cooperative effect on the on-rate that we uncovered has a smaller effect, in general, on the run length than the reduced interference or increased availability of motors but is still significant ($\approx$10%), especially at low ATP and/or high motor numbers (see S17 Fig). While the cooperative increase in on-rates works for both rigid and fluid cargo, the effect is enhanced when more motors are available due to fluidity. Finally, we showed that increased fluidity results in an expected increased recruitment of motors to the microtubule, thereby increasing the number of bound motors, with the effect only being significant at very high motor density or low ATP. We found that the confluence of decreased interference, decreased off-rate, increased on-rate and increased motor recruitment can lead to positive feedback resulting in dramatic effects on the run length, under the right conditions. We predict, for example, that run length is insensitive to cargo surface fluidity for moderate numbers of motors at saturating ATP but can increase by *several fold* for fluid membranes at low ATP conditions. Taken together, our work reconciles varying experimental results including the observed insensitivity of run length to cargo fluidity at moderate motor densities and high ATP concentrations [44, 45]. Our prediction of a significant enhancement of run length at low ATP is potentially physiologically important as an adaptive mechanism, *in vivo*, under stress or starvation conditions [62].

We developed a generalized version of the analytical expression for run length that accounts for cargo surface fluidity and expect it to be useful to explore a wide range of parameter spaces for different motor types and cargo geometries going beyond our simulations, which were mostly done with a fixed cargo radius of 250 nm. Cargo geometry enters into the effective on-rate for motors which scales as $\pi_{ad} = \pi_0 S_a/(S_I + S_a)$. The increase in access area, $S_a$, as the cargo is pulled closer to the MT is determined by the curvature of the cargo surface and sets the magnitude of the cooperative increase in motor numbers. Even considering a fixed cargo height and high $D$, the on-rate still depends on the ratio of access area to the total surface area which

decreases with increasing radius of curvature of the cargo. Thus, from our generalized version of Eq 1, we expect an increased on-rate and higher run lengths for smaller cargo with a fixed number of motors (see S18 Fig). This is consistent with our simulation results (S21 Fig) and with other numerical studies [36]. The smallest cargos, with sizes comparable to synaptic vesicles of about 100 nm [63], in fact, show an enormous increase in the run length over a range of fluidity that is consistent with the ranges for *in vivo* membranes in different contexts [40, 64]. For synaptic vesicles with fluid membranes therefore, transport over very large axonal distances, is a natural outcome of our model. If the surface density of motors is fixed, on the other hand, the quadratic increase in the number of available motors compensates for the reduction in on-rate resulting in larger run lengths for larger cargo (S19 Fig). In general, the changes in run length over physiological ranges of fluidity imply that surface fluidity could be used as a control parameter to regulate transport.

We also explored the effects of load on motor team performance and showed that motors on lipid cargo can undergo dynamic clustering and enhanced collective force generation. *In vivo*, increased load may be due to trapping or steric occlusion of the cargo, in which case motors can potentially cluster together exerting increased directed force against the load, perhaps even freeing the cargo. Such an adaptive response involving motor clustering has been observed in axonemal endosomes [56] and increased collective force generation in response to increased load could potentially be important in maintaining transport *in vivo* where crowding is high.

Our results on the transport properties of fluid cargo over a wide range of surface fluidity, ATP concentration, motor number and cargo size offer quantitative guidelines for the design of artificial cargo driven by kinesin teams. While all our results are specifically using parameters for kinesin motors, they can be readily generalized to other systems with different geometries of cargos with different motors coupled by a fluid surface. Thus our results could be of general value in designing artificial cargo utilizing teams of engineered versions of non-cooperative motors with fluid coupling to enhance load-sharing and processivity. More generally, coupling intrinsically non-cooperative mechanical force-generating elements via a fluid surface could be a generic mechanism to promote load-sharing that is exploited within the cell.

## Materials and methods

We developed a Brownian dynamics model to address the question of how cargo surface fluidity influences transport by teams of kinesin motors. Early models of multi-motor cargo transport used a mean-field approach where all motors were considered to share the load equally [65]. Later stochastic models that assume unequal load sharing [30, 43, 46, 52, 66–68] were found to describe experimental observations better [46, 69, 70]. The next improvement to the model was to explicitly consider that motors are bound to a three dimensional spherical cargo surface, that was typically considered rigid [51, 71]. Only a few recent models have attempted to consider the lipid membrane on the cargo surface [36, 45, 54, 56]. For example, Lombardo *et. al.* [54], in the context of transport of vesicles by teams of myosins, modeled an ideally fluid cargo surface where motors instantaneously relax the tangential component of force. Our model is closer to that of Bovyn et. al. [36] where we consider unequal load sharing with explicit implementation of motor diffusion on the cargo surface.

In our model, we consider a spherical cargo of radius, $R$ ($R = 250$ nm in most cases) which is decorated with a given number of molecular motors ($N$) on the surface. $N$ doesn't change during the cargo run, meaning there is no binding and unbinding of motors between the cargo surface and the solution. Each of these $N$ motors is initially assigned a random, uniformly distributed anchor position on the cargo surface. Molecular motors on rigid cargo are

fixed on the cargo surface. So a given initial configuration of motors on the rigid cargo surface persists throughout the cargo run whereas motors on lipid cargo diffuse on the surface.

In this study we considered that all $N$ motors are kinesin motors which have a rest length of $L_{mot}$ = 57 nm [53]. We assume that an unbound kinesin motor binds to the microtubule with constant rate, $\pi_0$ = 5 s$^{-1}$ [43], if some part of the microtubule is within $L_{mot}$ distance from the anchor position of that unbound motor. In other words, unbound motors bind with a specific rate to the microtubule if they can access the microtubule. The number of motors on the cargo, ($N$) is assumed to be constant, similar to several other modeling studies [36, 72]. In other words, we assume the timescale for motor detachment from the cargo is much larger than the lifetime of the cargo on the microtubule.

Each cargo run is initiated with at least one motor bound to microtubule and stopped when all the motors detach from the microtubule. This procedure is similar to Bovyn. *et. al.* [36] allowing a direct comparision between the runlength results between our studies. However, in other models [72] the cargo is tracked even during bulk diffusion when all motors are unbound till some motor binds back to the microtubule.

A microtubule bound motor is characterized by two position vectors, anchor point ($\vec{A}$) on the cargo surface and head ($\vec{H}$) on the microtubule. The head position is defined just by its continuous position along the x-axis, i.e. we do not model the lattice structure of the microtubule unlike in a recent Brownian dynamics model [73]. We justify this based on the fact that the microtubule has multiple protofilaments and when two kinesins are on different protofilaments they can have same x-position on the microtubule. There are also models that have explicitly incorporated the fact that the motor heads cannot step on each other [54, 73], which is likely of more importance when the filament has only a few tracks like actin, which has only two protofilaments in a helix, compared to 10–15 for microtubules.

A microtubule bound motor is assumed to exert a spring-like force when the length of motor exceeds the motor's rest length ($L_{mot}$) with a force constant, $k_{mot}$ = 0.32 pN/nm [74, 75]. Let $\vec{L} = \vec{A} - \vec{H}$ and $L = |\vec{A} - \vec{H}|$. The force exerted by the motor on the cargo is then given by,

$$\vec{F} \quad = -k_{mot}(L - L_{mot})\hat{L} \quad L > L_{mot} \tag{2}$$

$$= 0 \qquad\qquad L \leq L_{mot} \tag{3}$$

The translation velocity of the center of mass of cargo is given by the (overdamped) Langevin equation

$$\frac{d\vec{X}}{dt} = \frac{1}{\gamma_c}\left[\sum_{j=1}^{N}\vec{F}_j + \vec{F}_{steric}\right] + \vec{\zeta} \tag{4}$$

Here, $\vec{F}_j$ is the force exerted on cargo by the j$^{th}$ motor. $\vec{F}_{steric}$ is the spring-like steric force on the cargo from the microtubule, represented with a high force constant $10k_{mot}$. We note that if the vesicle is deformable the steric spring constant could be significantly smaller. This force is present only if the cargo-microtubule distance is less than the sum of the cargo and microtubule radii. $\vec{\zeta}$ is the random force on the cargo due to collisions with the intracellular medium. We assume a normally distributed noise with zero mean, $\langle\vec{\zeta}\rangle = 0$ and the fluctuation-dissipation relation, $\langle\zeta_\mu(t)\zeta_\nu(t')\rangle = 2\gamma_c k_B T\delta_{\mu\nu}\delta(t - t')$. $\gamma_c$ is the friction co-efficient for the cargo given by $\gamma_c = 18\pi\eta_\nu R$, where $\eta_\nu$ is the co-efficient of viscosity of cytoplasm experienced by cargo. We approximated $\eta_\nu$ to be equal to the co-efficient of viscosity of water in room temperature

$(20°C)$, $\eta_\nu = 10^{-3}$ Pa.s. We integrate Eq 4 using the Euler-Maruyama scheme

$$\vec{X}(t + \Delta t) = \vec{X}(t) + \frac{\Delta t}{\gamma_c}\left[\sum_{j=1}^{N}\vec{F}_j + \vec{F}_{steric}\right] + \sqrt{2\gamma_c k_B T \Delta t}\vec{\xi} \tag{5}$$

where $\vec{\xi} = (\xi_1, \xi_2, \xi_3)$ and $\xi_1, \xi_2, \xi_3$ are drawn from normal distribution with zero mean and unit variance.

In addition to translational motion, the cargo also has rotational dynamics due to thermal fluctuations and torques from motor forces. Consider a motor at position $\vec{A}_i$ exerting a force $\vec{F}_i$ on the cargo. The torque on the cargo due to this motor is $\vec{r}_i \times \vec{F}_i$. The total torque on the cargo due to all the motor forces is

$$\vec{\tau} = \sum_{i=1}^{N}\vec{r}_i \times \vec{F}_i \tag{6}$$

The angular displacement in time $\Delta t$ taking into account this torque and thermal fluctuations is

$$\Delta\vec{\theta} = \frac{\vec{\tau}}{\gamma_R}\Delta t + \alpha\sqrt{4D_R\Delta t}\,\hat{n} \tag{7}$$

Where, $\gamma_R$ is the friction co-efficient, given by $8\pi\eta_\nu R^3$. $D_R$ is the rotational diffusion constant of the cargo. $\alpha$ is a calibration constant to match the experimentally measured rotational mean square displacement. We set the rotational diffusion constant $(D_R)$ for lipid cargo to be equal to that of a free spherical bead in solution, which is $k_B T/\gamma_R = k_B T/8\pi\eta_\nu R^3$. The rotational diffusion constant for rigid cargo bound by one motor was measured [76] to be $7 \times 10^{-2}$ $rad^2$ $s^{-1}$. In our simulations we used this value to calibrate the rotational diffusion of rigid cargo (see S1 Appendix). $\hat{n}$ in Eq 7 is given by $\hat{n} = (n_1, n_2, n_3)$ which is a random unit vector in 3-dimensions. To get this, we draw numbers a and b from uniform distribution in the interval [0, 1]. Calculate angles $\theta_t = cos^{-1}(2a - 1)$ and $\phi_t = 2\pi b$. Then $\hat{n} = (n_1, n_2, n_3) = (sin\,\theta_t cos\,\phi_t, sin\,\theta_t sin\,\phi_t, cos\,\theta_t)$.

Let $\Delta\theta$ be the magnitude and $\hat{\omega} = (\omega_x, \omega_y, \omega_z)$ be the direction of this angular displacement vector $\Delta\vec{\theta}$. The Rodrigues' rotation matrix corresponding to this rotation is [77]

$$R_{\hat{\omega}}(\Delta\theta) = \mathbb{I} + \tilde{\omega}\sin\Delta\theta + \tilde{\omega}^2(1 - \cos\Delta\theta) \tag{8}$$

where $\mathbb{I}$ is the $3 \times 3$ identity matrix and $\tilde{\omega}$ is given by

$$\tilde{\omega} = \begin{bmatrix} 0 & -\omega_z & \omega_y \\ \omega_z & 0 & -\omega_x \\ -\omega_y & \omega_x & 0 \end{bmatrix} \tag{9}$$

We update each anchor point position using this matrix

$$\vec{A}_i = \vec{X} + R_{\hat{\omega}}(\Delta\theta)(A_i - \vec{X}) \tag{10}$$

At each time step we also update the anchor positions of each motor on the lipid cargo surface using a similar Brownian dynamics formalism given by

$$\vec{A}(t + \Delta t) = \vec{A}(t) + \Delta l_\theta\hat{\theta} + \Delta l_\phi\hat{\phi} \tag{11}$$

where $\Delta l_\theta$ and $\Delta l_\phi$ are the small displacements along $\hat{\theta}$ and $\hat{\phi}$ directions in the plane tangential

to cargo surface at $\vec{A}(t)$. $\Delta l_\theta$ and $\Delta l_\phi$ are given by

$$\begin{pmatrix} \Delta l_\theta \\ \Delta l_\phi \end{pmatrix} = \sqrt{2D\Delta t}\begin{pmatrix} \xi_a \\ \xi_b \end{pmatrix} + \frac{\Delta t}{\gamma_s}\begin{pmatrix} F_\theta \\ F_\phi \end{pmatrix} \tag{12}$$

where $\xi_a$ and $\xi_b$ are random variables obtained from normal distribution with zero mean and unit variance. $D$ is the diffusion constant for motor diffusion on cargo surface and $\gamma_s$ is the friction coefficient given by $\gamma_s = k_B T/D$. $(F_\theta, F_\phi)$ are the components of motor forces along $\hat{\theta}$ and $\hat{\phi}$ respectively which can be obtained from the motor force in Cartesian co-ordinates, $\vec{F} = (F_x, F_y, F_z)$ as follows

$$\begin{pmatrix} F_\theta \\ F_\phi \end{pmatrix} = \begin{pmatrix} \cos\theta\cos\phi & \cos\theta\sin\phi & -\sin\theta \\ -\sin\phi & \cos\phi & 0 \end{pmatrix}\begin{pmatrix} F_x \\ F_y \\ F_z \end{pmatrix} \tag{13}$$

At every time step, each bound kinesin motor hydrolyses an ATP molecule with certain probability and attempts to move forward on the microtubule. This stepping probability is a function of the motor force and also the ATP concentration. We have adopted the following relation for the stepping probability [47]

$$p_{step}\big([\text{ATP}], \vec{F}\big) = 1 - e^{-\nu\Delta t/\delta} \tag{14}$$

where $\delta$ is the step size, the distance moved by motor after hydrolyzing one ATP molecule ($\delta = 8$ nm [13, 78, 79]). $\nu$ is the velocity of kinesin motor which is a function of the ATP concentration and motor force and is taken to be

$$\nu\big([\text{ATP}], \vec{F}\big) = \nu_0\big([\text{ATP}]\big)\tilde{\nu}\big(\vec{F}\big) \tag{15}$$

where $\nu_0\big([\text{ATP}]\big)$ is the velocity of motor under no-load condition at a given ATP concentration. The ATP dependence of $\nu_0$ is described by the Michaelis-Menten equation

$$\nu_0\big([\text{ATP}]\big) = \frac{\nu_{max}[ATP]}{K_m + [ATP]} \tag{16}$$

We considered the no-load velocity at saturated ATP concentration to be $\nu_{max} = 800$ nm/s [47, 48] and $K_m = 44\,\mu$M [80].

$\tilde{\nu}(\vec{F})$ gives the force dependence of the velocity. In the hindering direction, we assume [46, 47]

$$\begin{aligned} \tilde{\nu}_{hind}(\vec{F}) \quad &= \left[1 - \left(\frac{F}{F_s}\right)^w\right] \qquad F < F_s \\ &= 0 \qquad\qquad\qquad\quad F \geq F_s \end{aligned} \tag{17}$$

F is the magnitude of motor force, $F = |\vec{F}|$. $F_s$ is the stall force, the value of force beyond which kinesin motor stops walking. We considered $F_s = 7$ pN [48, 81, 82] and $w = 2$ [69]. In the assistive direction, velocity is assumed to be independent of force magnitude, $\tilde{\nu}_{asst}(\vec{F}) = 1$ [47, 48].

Experimentally it is found that a kinesin motor is more likely to detach from the microtubule when one head is detached from the microtubule while trying to take a step than when both the heads are bound to the microtubule [80, 83, 84]. In our model we assume that a

motor can detach only when it tries to take a step. At every time step we first check whether a bound motor tries to make a step with probability $p_{step}([ATP], \vec{F})$ using Eq 14. If it tries to take a step, we check whether it detaches from the microtubule before completing the step using a microscopic off-rate whose value is calibrated based on the experimentally observed off-rate as a function of force, $F$, at saturating ATP concentrations.

$$\epsilon_{micro}(\vec{F}) = \frac{\epsilon_{obs}(\vec{F})}{p_{step}([ATP]= 2\,mM, \vec{F})} \qquad (18)$$

$p_{step}([ATP]= 2\,mM, \vec{F})$ is the probability to step forward in time step $\Delta t$ at high ATP concentration of 2 mM.

For hindering forces, we used the following relationship between observed off-rate and magnitude of motor force $F$ developed in [15, 85, 86] based on Kramer's theory [87] and used in several studies [23, 46, 47]

$$\epsilon_{obs}^{hind}(\vec{F}) = \epsilon_0 e^{F/F_d} \qquad (19)$$

$\epsilon_0$ is the off-rate under no load condition. We used $\epsilon_0 = 0.79\ s^{-1}$ [47, 48]. $F_d$ is the detachment force. We approximated $F_d$ to be equal to the stall force $F_s$. For assistive forces, the relationship between observed off-rate and magnitude of force is taken to be [47, 48]

$$\epsilon_{obs}^{asst}(\vec{F}) = \epsilon_0 + 1.56 \times 10^{12} F \qquad (20)$$

Our model could easily be extended to incorporate more detailed chemo-mechanical models of motors [72, 73].

The code for our model and data are available at https://github.com/nsarpangala/lipid-cargo-transport.

## Supporting information

**S1 Fig. Motor and cargo position and corresponding forces.** Variables and the simulation parameters same as Fig 2a.
(EPS)

**S2 Fig. The fraction of the force distribution with a magnitude of force *f* greater than or equal to 0.01$F_s$, as a function of the diffusion constant for the different number of bound motors n.** Error bars represent the standard error of the mean obtained by the bootstrap method. $F_s$ is the stall force of kinesin. The fraction increases as a function of *n* for rigid cargo but decreases for lipid cargo. Required force distributions as a function of *n* and *D* were obtained using the same procedure as explained in Fig 2b (simulations without rotational diffusion, N = 16, [ATP] = 2 mM).
(EPS)

**S3 Fig. Fraction of force distribution with force magnitude greater than 0.01$F_s$ in the hindering direction.** Required force distributions as a function of *n* and *D* were obtained using the same procedure as explained in Fig 2b (simulations without rotational diffusion, N = 16, [ATP] = 2 mM).
(EPS)

**S4 Fig. Fraction of force distribution with force magnitude greater than 0.01$F_s$ in the assistive direction.** Required force distributions as a function of *n* and *D* were obtained using the

same procedure as explained in Fig 2b (simulations without rotational diffusion, N = 16, [ATP] = 2 mM).
(EPS)

**S5 Fig. Mean value motor force (in Newtons) experienced by a bound motor as a function of the number of bound motors $n$ and diffusion constant $D$.** Mean force magnitude is negative which means on an average the motor is in the hindering direction, as one would expect because of active motion of motor on MT. The magnitude of mean force decreases with an increase in the number of bound motors due to load sharing. (simulations without rotational diffusion, N = 16, [ATP] = 2mM).
(EPS)

**S6 Fig. The mean variance of forces (in units of $N^2$) among bound motors.** Calculated as $\frac{1}{S}\sum_{i=1}^{S} \sigma_{|f|}^2(i)$ where the summation is over all the sample time points where the number of bound motors is $n$. $\sigma_{|f|}^2(i)$ is the variance in the magnitude of force experienced by the n bound motors at $i^{th}$ data sample. $S$ = 10000 except for (i) $D$ = 0 $n$ = 3, $S$ = 7360 (ii) $D$ = 0 $n$ = 4, $S$ = 1083 (iii) $D$ = 0.001 $n$ = 4, $S$ = 3754 (iv) $D$ = 0.01 $n$ = 4, $S$ = 6396 (v) $D$ = 0.01 $n$ = 4, $S$ = 3508 (vi) $D$ = 0.1 $n$ = 4, $S$ = 4883. (simulations without rotational diffusion, N = 16 [ATP] = 2 mM).
(EPS)

**S7 Fig. Simulation results for different cargo radii.** (a) Distribution of the absolute magnitude of motor forces in rigid and lipid cargoes for two different cargo radii, without considering rotational diffusion of cargo. (b) Distribution of the absolute magnitude of motor forces in rigid and lipid cargoes for two different cargo radii, including rotational diffusion of cargo. (c) Mean force correlation between x-components of motor forces (d) Mean single motor off-rate. N = 16, [ATP] = 2 mM.
(EPS)

**S8 Fig. Distribution of cargo displacements (position fluctuations) measured in 0.01 s time intervals along (a) x-axis (b) y-axis and the (c) z-axis.** In general, fluctuations increase with the increase in the fluidity of the cargo surface. Vertical lines indicate the mean values of fluctuations, interpreted as $v_{eff}\delta t$ where $v_{eff}$ is the effective velocity of cargo and $\delta t$ is the time interval used for measuring fluctuations (0.01 s). Fluctuations along the x-axis have a positive mean ($v_{eff}\delta t \approx 8nm$) because of the active motion due to molecular motors. Fluctuations along the y-axis and z-axis have a 0 mean. Fluctuations along the z-axis are much narrower than fluctuations along the y-axis and x-axis because of the steric force due to the microtubule. Data were obtained from the transport of cargoes (without rotational diffusion) with a total of $N$ = 16 motors at [ATP] = 2 mM. 200 cargo runs were considered for each diffusion constant.
(EPS)

**S9 Fig. Standard deviation (nm) of the position fluctuations as a function of diffusivity and the number of bound motors.** Data were obtained from the simulation of the transport of cargoes (without rotational diffusion) with a total of $N$ = 16 motors at [ATP] = 2 mM. 200 cargo runs were considered for each diffusion constant.
(EPS)

**S10 Fig. (a) Conditional on-rate measured from the simulation trajectories**. We simulated 200 cargo runs (without rotational diffusion) for N = 16 [ATP] = 2 mM case and recorded the value of the number of bound motors at a sampling rate of 100 s$^{-1}$. With this data, we identified all the $n$ bound motors to $n$ + 1 bound motor transitions, calculated the mean time for such transitions, and then the mean rate (1/meantime). We then divided this rate by the

number of unbound motors $N − n$ to obtain the conditional on-rate per motor. We then repeated the analysis process for a different number of bound motors $n$ and diffusion constants $D$. **(b) Comparing the measured conditional on-rate with analytical expressions**. Black triangles with dashed lines provide conditional on-rate where we consider the increase in on-rate of a single motor with an increase in the number of bound motors. Grey circles with a dashed line provide conditional on rate assuming a constant single motor on-rate. It can be inferred that if we measure this quantity in experiments one can expect to see only a slight increase as the cargo comes closer to the microtubule. This is because as the number of bound motors increases, the effective detachment rate of the n-bound state increases, hence the gating time decreases. Our analysis shows that the decrease in gating time contributes more to the increase in the conditional on-rate than the increase in the single motor on-rate. See S2 Appendix for more details.
(EPS)

**S11 Fig. Ensemble average (± SEM) of the number of bound motors for three different cargo-motor systems.** Vertical lines indicate the estimated time-scales, mean time for a new motor to bind—$\tau_{bind}$ (dashed lines) and mean unbinding time of a kinesin motor—$\tau_{off}$ (solid black line). N is the total number of motors on cargo, High [ATP] = 2 mM, Low [ATP] = 4.9 $\mu$M. The averaging was performed over 200 cargo runs (without rotational diffusion) each. Please refer to Table 1 in S2 Appendix for more information on $\tau_{bind}$ and $\tau_{off}$.
(EPS)

**S12 Fig. The distribution of the angular distance (ΔΘ) between the anchor positions of microtubule bound motors along the great circles connecting them (ΔΘ).** 200 cargo runs were considered for each parameter set with data sampling rate = 100 s$^{-1}$).
(EPS)

**S13 Fig. (a) Average number of motors (± SEM) (b) Runlength (± SEM) for more values of diffusion constants.** Averaging was performed over 200 cargo runs in each case.
(EPS)

**S14 Fig. Probability distribution of the number of bound motors in three different cargo-motor systems. High [ATP] = 2 mM, Low [ATP] = 4.9 $\mu$M.** Data were obtained from 200 cargo runs for each case.
(EPS)

**S15 Fig. Velocity of cargo along x-axis, $v_x^{eff}$ as a function of the number of bound motors and diffusion constant.** Velocity is measured as the ratio of mean displacement along x-axis in a given time window (we took $\delta t = 0.1s$) to $\delta t$. Cargo position data was obtained from the simulations of the transport of cargoes with a total of $N = 16$ motors at [ATP] = 2 mM. 200 cargo runs were considered each for each diffusion constant. We recoreded data at a sampling rate of 100 $s^{-1}$. Data in (b) is for N = 16, [ATP] = 2 mM without rotational diffusion. Error bars represent the standard error of the mean. For figure (a) we considered 10000 random time windows over 200 cargo runs to get the mean displacement for each diffusion constant. For figure (b) we considered all the time window samples with given $n$ over all the cargo runs.
(EPS)

**S16 Fig. Average value of motor off-rate as a function of fluidity of cargo surface for N = 4 [ATP] = 4.9 $\mu$M.** (Sample size = 530 from 200 cargo runs).
(EPS)

**S17 Fig. Comparison between run lengths from our simulations (circles) and analytical estimates (maroon triangle, solid, dashed and dash-dotted lines) as described in the main text.**
(EPS)

**S18 Fig. Analytically estimated runlengths for different cargo radii for a fixed number of motors on cargo.** The general method for calculating runlength analytically is described in the main text. We have numerically computed the access area, $S_a$, considering the typical distance between cargo surface and MT for 1 motor bound case. $\tau_{off}^m$ for computing influx area, $S_I(D) = \sqrt{2D\tau_{off}^m}$, was taken to be 1 s which is the typical motor unbinding time at saturating ATP concentration. Yellow vertical bands in (a) and (b) correspond to the range of physiologically relevant diffusion constants of motors.
(EPS)

**S19 Fig. Analytically estimated runlengths as a function of cargo radius (R) for a fixed surface motor density ($\sigma$).** The general method for calculating runlength analytically is described in the main text. We have numerically computed access area, $S_a$, considering the typical distance between cargo surface and MT for 1 motor bound case. $\tau_{off}^m$ for computing influx area, $S_I(D) = \sqrt{2D\tau_{off}^m}$, was taken to be 1 s which is the typical motor unbinding time at saturating ATP concentration. Yellow vertical bands in (a) and (b) correspond to the range of physiologically relevant diffusion constants of motors.
(EPS)

**S20 Fig. Comparison of the number of bound motors and cargo runlength without and with rotational diffusion in the model.** (a) Average number of bound motors (b) Runlength. Error bars represent the standard error of the mean. [ATP] = 2 mM.
(EPS)

**S21 Fig. Simulation results for different cargo radii.** Runlength, and average number of bound motors in rigid and lipid cargies as a function of the cargo radius. N = 16, [ATP] = 2 mM.
(EPS)

**S22 Fig. Simulation results for different motor stiffness ($k_{mot}$).** Runlength, average number of motors and motor off-rate for different motor stiffness. N = 16 [ATP] = 2 mM. Darkgreen is lipid cargo ($D = 1\ \mu m^2\ s^{-1}$) and grey is rigid cargo ($D = 0$).
(EPS)

**S23 Fig. The flow chart of the simulation.** The cargo is held near the microtubule until at least one motor is bound. Once a motor is bound, we loop over the series of steps shown in the middle until either all motors are unbound or maximum time is reached. At regular time intervals, we record relevant data like the cargo center of mass, anchor positions of motors on cargo surface, head positions of bound motors on the microtubule.
(EPS)

**S1 Video. Visualization of our cargo transport model.** The light brown colored sphere represents the cargo. Small red spheres are unbound motors. Unbound motors diffuse on the cargo surface and bind to the microtubule with a rate of 5 s$^{-1}$ when they come closer than their rest length to the microtubule. Blue sticks represent bound motors that walk along the microtubule with a rate dependent on the force that they experience. Bound motors unbind with a force-dependent rate. The microtubule is represented as the dark-green horizontal line. Data for this movie was obtained from the simulation of a cargo with 16 motors. The data sampling rate

was $10^5$ s$^{-1}$. We used Mayavi [88] for visualization.
(MOV)

**S2 Video. Cargo dynamics viewed at a shorter time and length scale.** The data for this movie is the same as S1 Video.
(MOV)

**S1 Appendix. Notes on deformation of vesicle due to motor forces and rotational diffusion of cargo.**
(PDF)

**S2 Appendix. Estimation of time scales $\tau_{bind}$ and $\tau_{off}$ and a note on conditional on-rate.**
(PDF)

**S3 Appendix. Analytical estimation of run length of a rigid cargo.**
(PDF)

**S4 Appendix. Discussion on how the runlength of cargo can be different even though the average number of motors are the same.**
(PDF)

**S5 Appendix. Details of different force related metrics used in Fig 2.**
(PDF)

## Acknowledgments

We are grateful for helpful discussions with David Quint and Jing Xu and insightful comments and suggestions from referees.

## Author Contributions

**Conceptualization:** Niranjan Sarpangala, Ajay Gopinathan.

**Formal analysis:** Niranjan Sarpangala, Ajay Gopinathan.

**Funding acquisition:** Ajay Gopinathan.

**Investigation:** Niranjan Sarpangala, Ajay Gopinathan.

**Methodology:** Niranjan Sarpangala, Ajay Gopinathan.

**Project administration:** Ajay Gopinathan.

**Resources:** Ajay Gopinathan.

**Software:** Niranjan Sarpangala.

**Supervision:** Ajay Gopinathan.

**Validation:** Niranjan Sarpangala, Ajay Gopinathan.

**Visualization:** Niranjan Sarpangala, Ajay Gopinathan.

**Writing – original draft:** Niranjan Sarpangala, Ajay Gopinathan.

**Writing – review & editing:** Niranjan Sarpangala, Ajay Gopinathan.

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
