## [Decision Letter · Decision Letter 0]

25 Oct 2021

Dear Prof. Gopinathan,

Thank you very much for submitting your manuscript "Cargo-mediated mechanisms reduce inter-motor mechanical interference, promote load-sharing and enhance processivity in teams of molecular motors" for consideration at PLOS Computational Biology.

As with all papers reviewed by the journal, your manuscript was reviewed by members of the editorial board and by several independent reviewers. In light of the reviews (below this email), we would like to invite the resubmission of a significantly-revised version that takes into account the reviewers' comments.

In particular, reviewer #2 notes that the model is very similar to modelling work by Bovyn et al., and also lists several other related papers that provide a relevant context. I would ask you to carefully compare you work with the published work and include a balanced comparison and description of the advance over the previous work. Then the reviewers note that they felt the paper was difficult to read, in particular the argumentation was hard to follow; i.e. some of the results were hard to follow without knowing the model equations in detail. Reviewer #3, and to certain extent also reviewer #2 notes a relatively "one sided view" on the positive effects of diffusion on cargo transport, whereas there are also conflicting observations in the literature. We would ask you to provide a more balanced view of the novel insights in light of alternative, published models and in light of previous observations that contradict your model. Please take these, and also the other remarks of the reviewers into account in your revised manuscript.

In particular because of the similarity with previous work, we cannot make any decision about publication until we have seen the revised manuscript and your response to the reviewers' comments. Your revised manuscript is also likely to be sent to reviewers for further evaluation.

Sincerely,

Roeland M.H. Merks, Ph.D

Associate Editor

PLOS Computational Biology

Jason Haugh

Deputy Editor

PLOS Computational Biology

Reviewer's Responses to Questions

**Comments to the Authors:**

Reviewer #1: Review attached as PDF

Reviewer #2: In this paper Sarpangala and Gopinathan present simulations of a Brownian Dynamics model of transport of fluid and rigid cargo by multiple Kinesin-I motors. They find that membrane fluidity decreases negative interference between multiple bound kinesin motors, and increases motor binding (under some conditions). Additionally, they find that, regardless of membrane fluidity, Kinesin binding to the microtubule increases the binding of subsequent Kinesin by pulling the cargo closer to the microtubule. They do some analysis to estimate the contribution of these effects, and determine that the decrease in negative interference is the dominant effect under typical in vivo conditions.

For the most part, the model seems reasonable (though I did have some questions about it, see below). I particularly liked the use of both simulation and analysis to address questions of interest. Although the paper would be stronger with experimental support of its predictions, similar models have been shown to be sufficiently predictive that I thought the results could stand on their own. I found the results interesting, however I was left with a few major concerns and questions. First, how does this model differ from previous models, and what unique insights does it provide? Second, I found the paper difficult to read because I could not understand the results until I knew what went in to the model. Third, though most of the modeling seemed reasonable, I was not convinced that cargo rotation could be reasonably ignored. It is the first of these concerns that, to me, is most important. I will explain each concern more fully, and then follow with some more minor concerns.

Concern 1: How does this model differ from previous models?

Overall, I felt the paper was too focused on the results of the modeling, and not enough on how the results fit into previous modeling work.

One major concern I had reading the paper is that the model seems very similar to the model presented in Bovyn et al. 2021. To be fair, this paper is cited (reference 36 of the main text). However, the similarities and differences between that model and the authors’ model was not discussed, nor were the similarities and differences between the results of that study and authors’ results. In my mind, that’s the purpose, in part, of a Discussion — i.e., to discuss how the current results relate to the literature.

Building off this, although the authors do cite a good selection of modeling literature, it is incomplete and misses some recent studies that consider transport by multiple kinesin motors (e.g. Chen, Nam, Epureanu, PRE, 2018 and 2020; Takshak, Mishra, Kulkarni and Kunwar, Physica A, 2017) multiple dynein motors (Chowdhary, Kaplan, Che, Cui, Biophys J, 2018) and multiple myosin motors (Lombardo et al., Nature Comm 2017, PNAS 2019). Some of these consider fluid liposomes (e.g. Chowdhary and Lombardo). Note that these are just references that I can think of now, and not an exhaustive list. As with Bovyn et al, the authors’ model should, I think, be contrasted with these and the results compared.

As an example of a comparison that should, I think, be made is the authors’s result of positive cooperativity — i.e., the increasing binding rate that occurs when kinesin motors bind — with negative cooperativity discussed in Lombardo et al. 2017. In that paper, the authors’ model predicts that as myosin motors bind to actin, they restrict the movement of the cargo which then decreases the binding rate of subsequent motors. Moreover, they perform measurements in the optical trap to support this prediction. I believe the difference arises from different mechanics of the motors. In particular, kinesin are rather floppy while myosin are stiffer — however it might also arise from the fact that the authors neglect rotation of the cargo, and don’t include a torsional spring element that tends to keep the kinesin motors extending from the cargo in the normal direction.

Concern 2: The results are hard to understand without a description of the model.

While I think it’s ok to leave the details of the model to a methods section at the end of the paper, there must be some description of the model prior to the results. Without knowing the details of the mechanochemical cycle each kinesin undergoes (including how ATP fits in), the mechanical model of the kinesin molecules and other such details, it is impossible to understand where the results come from. As a particular example, the result of positive cooperativity arises from the assumption in the model that kinesin can bind to the microtubule if it is within a distance L_mot of the kinesin’s attachment to the cargo. As mentioned above, different assumptions about the mechanics of this connection can lead to negative cooperativity (e.g. Lombardo et al. 2017). Understanding how motors working together might increase velocity depends, at least in part, on the mechanochemical model of the motors (e.g. Nelson et al. 2014, PNAS, reference 42). Finally, I didn’t realize that cargo rotation was neglected in the model until I read the supplement.

I suggest that the model be described at the beginning of the Results. The precise details of the modeling and the equations might, perhaps, be saved for the methods section; however, there must be enough detail for the reader to understand and evaluate the rest of the Results section.

Concern 3:

I do not understand and am therefore not convinced by the authors arguments that rotational motion of the cargo may be neglected (see below):

In my reading of Gutierriez-Medina et al., it seemed to me that they measured a rotational diffusion constant of 7*10^-2 rad^2/s, a bit over an order of magnitude larger than that reported in Appendix S1. With this value, the time to rotate 90 degrees is 1 second, much more comparable to the stepping rate. In addition, they find that double-headed kinesin applies a significant torque to the bead, which restrains its motion. This is not included in the model.

In the same appendix, if I assume that the applied torque scales as f*R, I get 2.5*10^-19 Nm (1*10^-12*250*10^-9), in agreement with the authors. However, when I divide this value by 8*pi*eta_v*R^3 — which I calculate to be 8*pi*0.001*Pa*s*(250*10^-9 m)^3 = 3.8*10^-22 N*m*s, I get a rotational diffusion constant of order 10^4 — fifteen orders of magnitude larger than the value reported in Appendix S1. Intuitively, this makes some sense, since I would expect the bead to rotate as the motors step — indeed, such rotation was measured by Gutierrez-Medina et al, referenced above.

My understanding of Bovyn et al. 2021, is that they included rotational diffusion in their model, and found it important in their subsequent results.

More minor comments:

1. I did not like the authors stating that they had demonstrated various things “for the first time” — particularly given that they did not seem aware of some previous related work (see major comment 1).

2. I thought some of the claims in the abstract, particularly in the last two sentences, were not clearly backed up in the paper.

3. In the author summary, I thought there should be a sentence explaining why motors are non-cooperative. Otherwise, it seems like the authors are saying that the motors are non-cooperative and then explaining why they are, indeed, cooperative (or at least not non-cooperative, if you will pardon a double negative).

4. Is “increased mechanical efficiency,” referenced at the end of the introduction, defined and demonstrated?

5. Throughout the paper, the word “significantly” seems to be used to refer to things that appear different (i.e., “This distribution … is significantly broader …” bottom of page 3). I would use a different word, so it is not conflated with statistical significance.

6. Equation 14, the exponential dependence of reaction rate on force, is often attributed to Bell (GI Bell, Science, 1978). The connection between that and Kramers theory was known prior to Schnitzer et al 2000 (e.g. Evans and Ritchie, Biophysical Journal, 1997).

7. I found it confusing that the supplementary sections each had different reference numbering than the main text. Perhaps this is journal style and would be less confusing when the supplements are not in the same document as the main text; however, I lost some time looking through references from the main text instead of the appropriate reference from the supplement.

8. In appendix S3, it appears that reference 23 of the main text is cited, while all other references in the appendices are self-contained.

9. Overall, I thought the figures might communicate information a little bit more efficiently. For example, the same symbols/colors might be used to indicate the amount of cargo fluidity throughout the paper, symbols might be defined more clearly (e.g. in Fig. 3d, one has to read through the text to find out what the deltas represent).

10. It should be metioned that the viscosity of water used in the paper applies to room temperature (~20 C).

Reviewer #3: In this manuscript, the authors use Brownian dynamics simulations to explore transport by molecular motors that are attached to their cargo by fixed anchor points vs. by motors that can diffuse on a fluid cargo surface. Analyzing several complementary metrics such as force distributions and detachment rates of motors, they conclude that surface fluidity provides a simple mechanism to promote load sharing between individual motors, and thus leads to an increase in the number of active motors pulling on the cargo, as well as to increased run lengths of the cargo.

The study is well-performed and detailed, and the conclusions regarding the force sharing properties between motors are relevant and interesting for the general question of cargo transport by teams of motors. However, the manuscript in its current form provides a rather convoluted presentation of the main results, lacks relevant comparisons with experimental observations, and does not discuss comprehensively alternative scenarios that might influence the conclusions. I therefore think additional sets of simulations, a genuinely thorough restructuring of the text, and a corresponding revision of the figures are necessary to consider this study for publication in PLoS Computational Biology.

Below I list my major concerns:

M1) In general, physical mechanisms that impact the mechanical interference between motors are not discussed carefully and in sufficient detail: Basic aspects of load sharing & mechanical coupling between bound motors should be elucidated with further analysis from individual motor trajectories. For instance, what are typical separations between bound motors? How do these separations translate into force generation profiles? What are the typical timescales of force balance between motors? In the “Conclusion” the authors say “..lipid membrane which allows the attachment points to move and relax strain…”, but in fact the effect of a fluid membrane should be more of a delay in strain generation rather than relaxation of strain. Timescales of strain generation or force balance between bound motors for the two different cases of fluid vs rigid cargo could reveal further mechanisms underlying the changes e.g. in on/off rates or force generation profiles.

M2) Different motor stiffness regimes needs to be explored. The stiffness value used in this work (k_mot = 0.32pN/nm) is taken from a rather old study (Refs [59] and [60]). However, much smaller stiffness values within the range 0.05-0.2pN/nm were used in more recent studies such as in

Driver et al, PCCP 2010. These different stiffness regimes might markedly influence the off-rates or average run lengths of the cargo, e.g. as explored in Berger et al, PRL 2012.

M3) It is unclear to me why the authors only consider cargos of a single size (with radius R = 250nm). This is an important aspect that would presumably influence the attachment of and interference dynamics between motors. How do the main results such as the force distributions (Fig.2b), on & off rates of motors (Figs. 2e&g), as well as the mean run lengths (Fig.3c) depend on the cargo size, such as explored in Erickson et al PLoS Comp Biol 2011?

M4) Similar to M3, how would different parameter regimes of the motors influence the main results? This could be relevant to gain insights on distinct motor types such as myosins or other kinesins. For instance, would the load sharing properties change significantly if the motors had a small “no-load velocity” v_0 or a small stall force F_s? How would these different parameter sets influence the number of and the forces experienced by bound motors, such as explored in Ucar & Lipowsky, Nano Lett 2020?

M5) In general, the authors should try to include some experimental data to quantitatively test / compare their theoretical results. Otherwise their claim, in the abstract, “...th[u]s allowing us to reconcile different experimental results in different regimes” remains unsubstantiated. For example, it should be further elucidated with more targeted analysis and additional simulations / datasets from the literature why cargo velocities (Fig. S14) do not change significantly in contrast with previous experimental findings such as in Grover et al PNAS 2016.

Technical / minor comments:

m1) Fig. 3 is a rather arbitrary collection of different metrics, and it is very tiresome to extract a significant message out of it. In general, each item (i.e. Figure) in the manuscript should convey a single message. It might therefore be beneficial to include captions as single sentences for each figure.

m2) Fig.3(b): A schematic for the polar angle \\theta is necessary. Top panel: it is not very informative to plot constant probability values over a large set of \\theta values because the cargo rotations are neglected in simulations. Bottom panel: The message is presumably that the probability of motors accumulating at the bottom of the cargo increases over time but the plotting is very confusing. It might be better to complement these angles with distributions of anchor-anchor distances.

m3) Fig.3(d): This plot as well as the corresponding section (Fluid cargo have longer tun lengths than rigid cargo) in the main text are extremely tedious to read due to the many different quantities introduced in passing and it remains again unclear which message should be extracted from the different analytical estimates. This entire section and Fig.3, in my perspective, should be largely revised and rethought to clearly explain the important conclusions.

m4) Details are missing on how the explored metrics (such as forces acting on individual motors and force correlations) are calculated.

m5) In conclusion, the authors state “rigidly coupled motors experience…”, however, the motors are elastically coupled even in the case of a rigid cargo due to their elastic linkers. The terminology overall should be used more carefully.

**Have the authors made all data and (if applicable) computational code underlying the findings in their manuscript fully available?**

Reviewer #1: Yes

Reviewer #2: Yes

Reviewer #3: **No: **I cannot find a link to a public repository where the simulation code can be found.

PLOS authors have the option to publish the peer review history of their article (what does this mean?). If published, this will include your full peer review and any attached files.

Reviewer #1: **Yes: **Roop Mallik

Reviewer #2: No

Reviewer #3: No
---

## [Decision Letter · Decision Letter 1]

20 Apr 2022

Dear Prof. Gopinathan,

Thank you very much for submitting your manuscript "Cargo surface fluidity can reduce inter-motor mechanical interference, promote load-sharing and enhance processivity in teams of molecular motors" for consideration at PLOS Computational Biology. As with all papers reviewed by the journal, your manuscript was reviewed by members of the editorial board and by several independent reviewers. The reviewers appreciated the attention to an important topic. Based on the reviews, we are likely to accept this manuscript for publication, providing that you modify the manuscript according to the review recommendations.

Sincerely,

Roeland M.H. Merks, Ph.D

Associate Editor

PLOS Computational Biology

Jason Haugh

Deputy Editor

PLOS Computational Biology

[LINK]

Reviewer's Responses to Questions

**Comments to the Authors:**

Reviewer #1: The authors have done all that is reasonable to address my comments and comments of other reviewers. I am therefore in support of publication.

Reviewer #2: The authors have addressed most of my concerns except for:

1. (Line 337) The source of negative cooperativity in the model of Lombardo et al. 2017, according to my understanding of the paper, is not competition for binding sites. Instead, it is due to myosin binding restricting the motion of the cargo and thereby making it more energetically costly for subsequent motors to bind.

2. Along the same lines, actin is not a linear track (as stated a couple times in the paper, e.g. line 336, line 745). It has two protofilaments arranged in a helix.

3. While I appreciate the difficulty in writing a pithy and understandable significance statement, I do not think “unsynchronized” is the right word. It seems to me that the authors are trying to address the issue that processive motors work as a team in vivo, yet seem to work poorly as a team when coupled rigidly in vitro. They are adding to a growing body of literature that argues that membrane fluidity resolves this contradiction.

4. (Line 365) A reference to a Table is broken.

Reviewer #3: The substantial revisions of the authors have improved the manuscript to provide a more thorough analysis and a more balanced presentation and discussion of their modelling and results. I therefore recommend publication in general. However, I think some sections of the manuscript could benefit from improving the writing to increase the overall accessibility and thus eventual impact of the paper. This is most evident in the abstract, which is rather long and not quite eloquently written. Some suggestions (not to be taken literally but as guidelines) are:

(i) removing unncessary/uninformative words (e.g “simulation” instead of “computer simulation”, removing “canonically non-cooperative”).

(ii) replacing technical terms with more accesible alternatives (e.g. “..show that surface fluidity enhances mechanical load sharing between kinesins, which increases the average duration of motors on the filament and thus their collective processivity.” instead of “surface fluidity could lead to the reduction of negative mechanical interference between kinesins, enhancing load sharing thereby

decreasing single motor off-rates and increasing processivity.”)

(iii) replacing informal expressions (e.g. “transport distances” instead of “travel distances”)

**Have the authors made all data and (if applicable) computational code underlying the findings in their manuscript fully available?**

Reviewer #1: Yes

Reviewer #2: Yes

Reviewer #3: Yes

PLOS authors have the option to publish the peer review history of their article (what does this mean?). If published, this will include your full peer review and any attached files.

Reviewer #1: No

Reviewer #2: No

Reviewer #3: No

Figure Files:

Data Requirements:

Reproducibility:

References:

---

## [Editor Report · Decision Letter 2]

16 May 2022

Dear Prof. Gopinathan,

We are pleased to inform you that your manuscript 'Cargo surface fluidity can reduce inter-motor mechanical interference, promote load-sharing and enhance processivity in teams of molecular motors' has been provisionally accepted for publication in PLOS Computational Biology.

Best regards,

Roeland M.H. Merks, Ph.D

Associate Editor

PLOS Computational Biology

Jason Haugh

Deputy Editor

PLOS Computational Biology

---

## [Editor Report · Acceptance letter]

1 Jun 2022

PCOMPBIOL-D-21-01565R2 

Cargo surface fluidity can reduce inter-motor mechanical interference, promote load-sharing and enhance processivity in teams of molecular motors

Dear Dr Gopinathan,

I am pleased to inform you that your manuscript has been formally accepted for publication in PLOS Computational Biology. Your manuscript is now with our production department and you will be notified of the publication date in due course.

With kind regards,

Anita Estes
